# In-tube micro-pyramidal silicon nanopore for inertial-kinetic sensing of single molecules

Jianxin Yang[1], Tianle Pan[1], Zhenming Xie[1], Wu Yuan [1] ✉ & Ho-Pui Ho[1] ✉

Electrokinetic force has been the major choice for driving the translocation of molecules through a nanopore. However, the use of this approach is limited by an uncontrollable translocation speed, resulting in non-uniform conductance signals with low conformational sensitivity, which hinders the accurate discrimination of the molecules. Here, we show the use of inertial-kinetic translocation induced by spinning an in-tube micro-pyramidal silicon nanopore fabricated using photovoltaic electrochemical etch-stop technique for biomolecular sensing. By adjusting the kinetic properties of a funnel-shaped centrifugal force field while maintaining a counter-balanced state of electrophoretic and electroosmotic effect in the nanopore, we achieved regulated translocation of proteins and obtained stable signals of long and adjustable dwell times and high conformational sensitivity. Moreover, we demonstrated instantaneous sensing and discrimination of molecular conformations and longitudinal monitoring of molecular reactions and conformation changes by wirelessly measuring characteristic features in current blockade readouts using the in-tube nanopore device.

Nanopore sensing is a technique that involves monitoring temporal traces of current blockades as molecules pass through a nanopore driven by a force difference imposed across the pore[1–4]. This approach can easily be integrated into portable sensing devices with electronics[5] and has contributed substantially to many branches of the life sciences over the past two decades because of its application in nanopore sequencing[6,7]. Current nanopore devices primarily utilise electrokinetic force for single-molecule translocation to produce current blockade signals. However, this approach makes it challenging to control the translocation speed, resulting in non-uniform conductance signals with limited temporal resolution and therefore low conformational sensitivity for discriminating molecular features[8–14]. Although speed control with protein motors has been successfully demonstrated with biological nanopores, achieving a stable protein motor feed rate and high conductance drop as in solid-state nanopores remains challenging[15,16]. In addition, controlled molecular translocation in nanopores has been achieved using a nanopositioner or an

atomic force microscope cantilever[17–20]; however, this method requires tethered molecules, preventing complete translocation.

A nanopore with the desired shape and size and made of an appropriate material is imperative for the accurate detection and characterisation of molecules in question, such as proteins[21,22]. For on-film molecule sensing[23], a funnel-shaped nanopore plays an essential role in enabling targets distributed over a large range to better address and pass through the nanopores[24]. While nanopores fabricated in ultrathin nanometre-thickness films, such as lipid bilayers[25,26], $Si_3N_4$[27–30], $MoS_2$[31,32] and graphene[33,34], offer desirable chemical stability against analytes, silicon nanopores fabricated in micrometre-thickness wafers in a reproducible and scalable manner provide better mechanical strength to withstand high levels of inertial force[35]. Anisotropic chemical etching has emerged as a major technique for cost-effective fabrication of silicon nanopores[36,37]. However, this technique is suboptimal for fabricating silicon nanopores smaller than 8 nm owing to unavoidable over-etching[37–39].

[1]Department of Biomedical Engineering, The Chinese University of Hong Kong, Hong Kong SAR, China. ✉e-mail: wyuan@cuhk.edu.hk; aaron.ho@cuhk.edu.hk

Previous studies have demonstrated the fabrication of truncated pyramidal nanopores with sub-5-nm dimensions on silicon-on-insulator substrates using electron-beam lithography and less than 100-nm-thick membranes[40]. However, the Debye length of truncated pyramidal nanopores is less than the nanopore size and the electroosmotic effect dominates the molecule translocation[41], resulting in short and unstable dwell times in readings of inadequate conformationally sensitive signals for discriminating molecules[40–42]. Various approaches have been explored to extend the dwell times of molecules in nanopores, including elongating the sensing length of the nanopore[1,22,43,44], slowing the translocation speed by introducing new potential gradients[45–50], optimising test environments[51] and fine-tuning molecular surface charges[52,53]. However, the stochastic nature of molecular dynamics in nanopores influences the translocation speed and limits high-fidelity and sensitive measurements of molecular conformation. As a result, current investigations into molecular translocation characteristics have primarily focused on information extracted from the amplitude and duration of blockade signals[54,55].

Here, we present an in-tube sensing device that incorporates an inertial force kinetically regulated molecular translocation method into a micro-pyramidal silicon nanopore (MPSN) to achieve precise fabrication of sub-5 nm silicon nanopores and controlled single-molecule translocation. This MPSN is fabricated using photovoltaic electrochemical etch-stop technique that leverages the local photovoltage and its counter-etching effect (i.e., photoinhibition effect) induced by near-field optical irradiation and implements a high-precision electro-optical monitoring strategy. This wet-etching

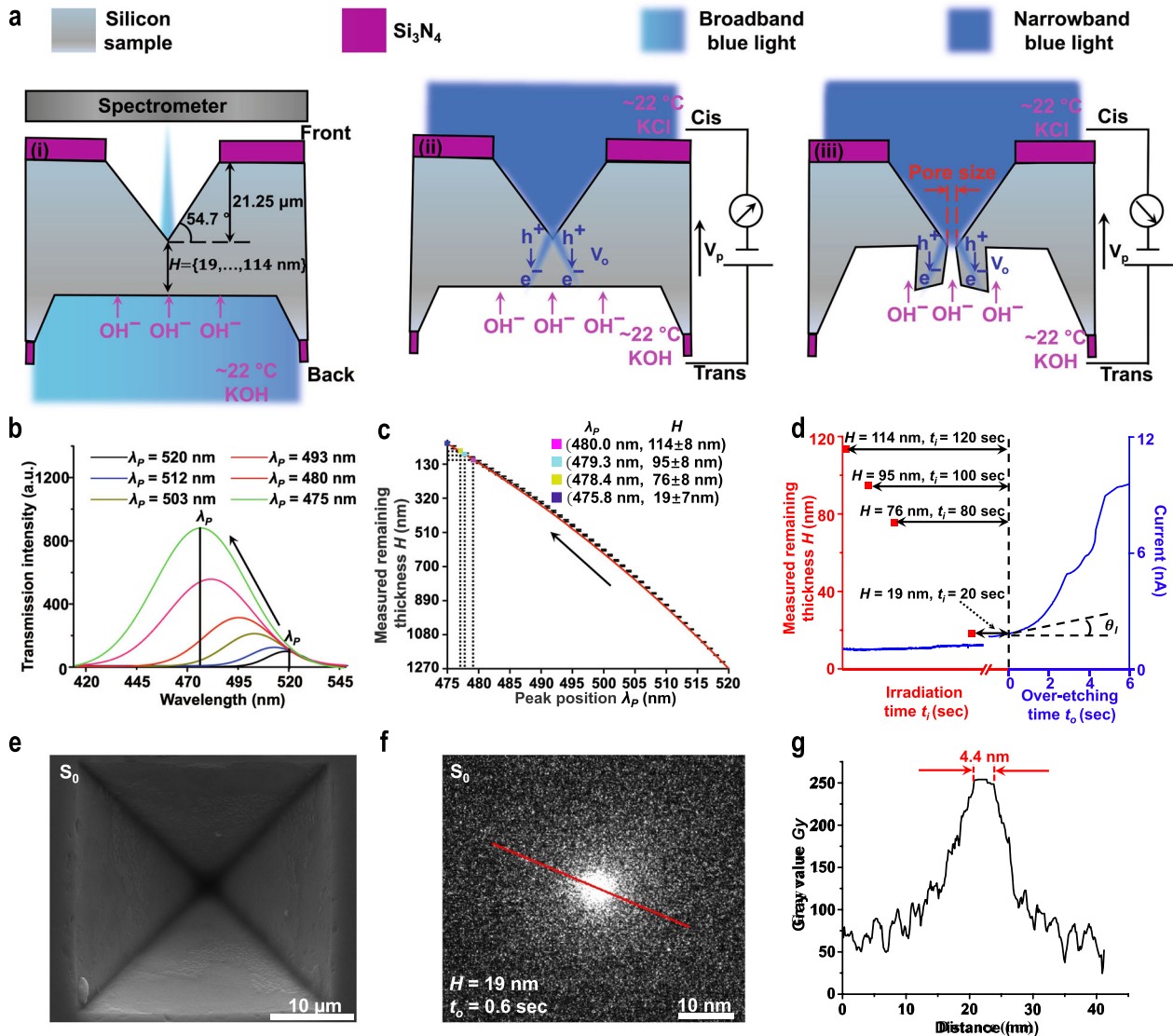

**Fig. 1 | Schematic of MPSN fabrication process, related control signals, and resultant MPSN. a** Pre-etching process using KOH to achieve remaining thickness $H$, i.e., distance between tip of micro-pyramid and silicon/etchant interface, ranging from 114 nm to 19 nm (i); local etching process using photoinhibition-assisted KOH etching to regulate etching rate and final pore size, i.e., the narrowest opening diameter, of MPSN (ii and iii). **b** Representative transmission spectra used to closely monitor remaining thickness during pre-etching process. Spectral peak $\lambda_p$ of 520 nm corresponds to $H$ of 1270 nm. **c** The spectral peak was measured using a spectrometer, while the remaining silicon thickness was measured using a contact profilometer. The black dots represent experimental data from 4 samples, and the red line represents the fitting result. Based on four wafer tests, the standard deviation of the measured remaining silicon thickness at the tested spectral peak is approximately 8 nm. **d** Current signal ($I_P$, blue line) measured during local etching process. Depending on remaining thickness $H$, different irradiation times $t_i$ are needed to achieve silicon perforation, leading to over-etching stage signified by current ramp with slope $tan\theta_I$ over 0.5 nA/sec. Pore size of MPSN is then precisely controlled by over-etching time $t_o$. Front-view SEM image (**e**) and back-view TEM image (**f**) of small MPSN $S_0$ with pore size of 4.4 nm fabricated using remaining thickness of 19 nm, irradiation time of 20 s, and over-etching time of 0.6 s. Pore size is further characterised using image grayscale analysis (**g**) of red line in **f**.

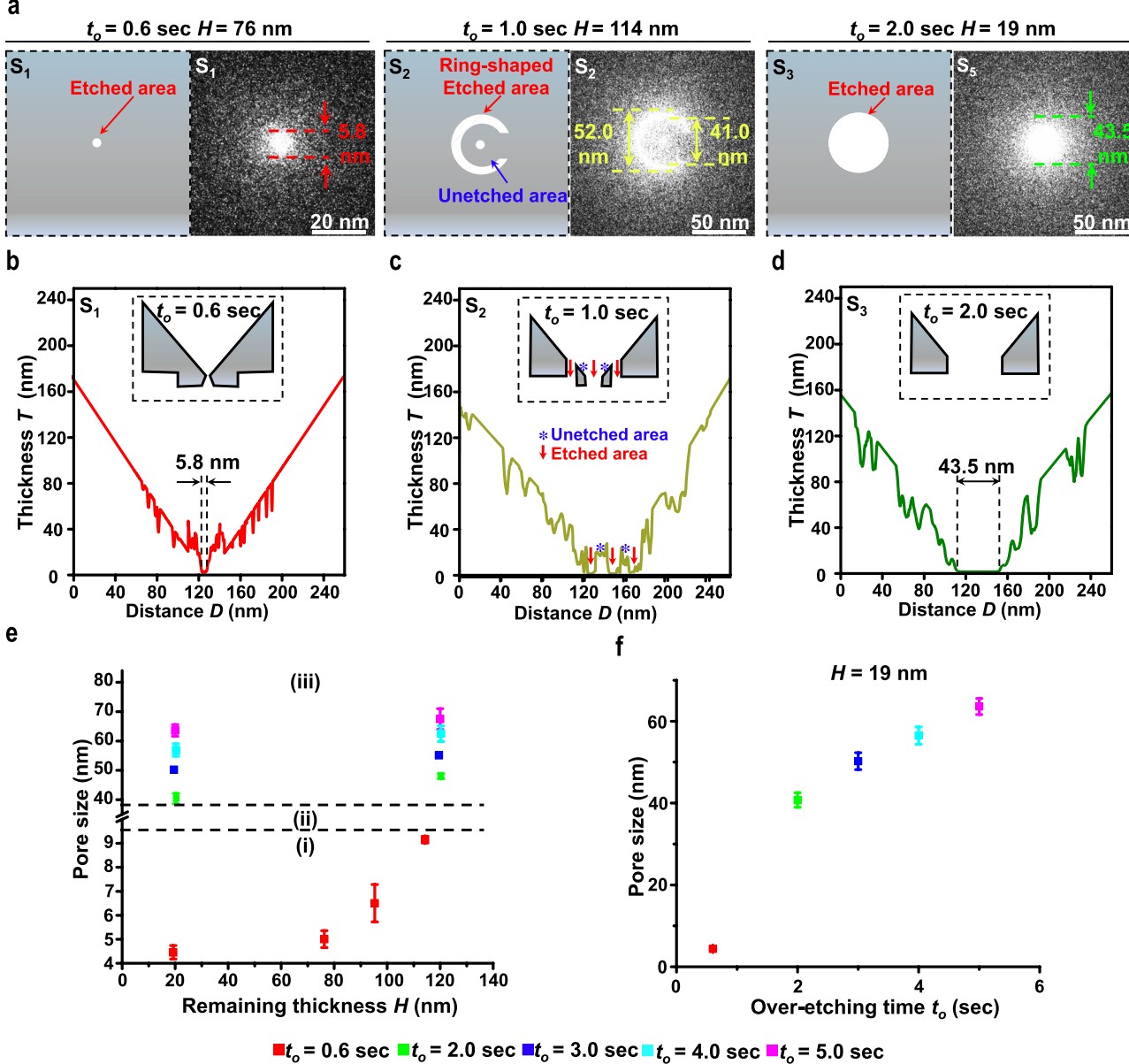

**Fig. 2 | TEM images and pore-size estimation of fabricated MPSNs. a** Schematic back-view nanopore shapes (left) and corresponding TEM images (right) of MPSNs fabricated with different remaining thickness $H$ (or irradiation time $t_i$) and over-etching time $t_o$. Side-view nanopore profiles of MPSN S1 (**b**), S2 (**c**), and S3 (**d**) derived from grayscale analysis. Insets (black boxes) show schematic side-view nanopore profile of each MPSN. **e** Pore sizes versus combinations of $H$ (or $t_i$) and $t_o$, indicating three working regions for fabricating different pore sizes: sub-10 nm (i), between 10 to 40 nm (ii), and over-40 nm (iii). **f** Increase in pore size with over-etching time $t_o$ when $H = 19$ nm (or $t_i = 20$ s). Error bars represent the standard deviations of experimental data measured from 3 samples.

technique enables a controllable, self-limited, and closed-loop chemical etching process capable of reliably and cost-effectively fabricating silicon nanopores with resolutions ranging from 65 nm to 4 nm. The resulting nanopores have a thickness of 21.25 μm, providing excellent structural strength to withstand high inertial force (see Supplementary Note 1, 2). Furthermore, an in-tube MPSN device was developed to enable inertial-kinetic translocation by decoupling molecular translocation from the electrokinetic effect and sensing proteins through the wireless measurement of current blockade signals at temporal resolution of at least 80 microseconds (see Supplementary Note 3). This device offers an adjustable dwell time of up to 100 ms and high conformational sensitivity for sensing proteins by kinetically regulating a funnel-shaped centrifugal force while maintaining electrophoretic and electroosmotic forces in a counterbalanced state within the nanopore. To validate our device, we demonstrated the real-time discrimination of the mass and shape of protein molecules as well as longitudinal monitoring of molecular reactions and conformational changes by quantifying the dwell times and detecting characteristic features in current blockade traces. The reliable fabrication of sub-5-nm silicon nanopores combined with the enhanced controllability offered by the inertial-kinetic translocation technique suggests promising potential for accurate and efficient molecule fingerprinting using the 'MPSN-in-a-tube' scheme.

## Results

### Fabrication of MPSNs using photovoltaic electrochemical etch-stop technique

The fabrication of MPSNs primarily involves the preparation process and pre-etching process followed by a local etching process (see Fig. 1a and Supplementary Note 4). During the preparation process, two 200-

nm thick $Si_3N_4$ layers were deposited on both sides of a (1-0-0) silicon wafer. Square patterns were then transferred onto the photoresist layers on silicon using photolithography. After removing the exposed $Si_3N_4$ layers through plasma etching, an inverted micro-pyramid structure with a depth of approximately 21.25 μm was fabricated on the front side of silicon using 30% (w/w) KOH etchant heated to 80.0 °C (see Methods).

In the pre-etching process, first 80.0 °C and then the room temperature (-22.0 °C) KOH etchant was used to etch silicon from the backside to achieve a remaining thickness of 19–114 nm. A spectral detection system consisting of broadband blue light and a spectrometer were used to precisely track the transmission spectral characteristics of the remaining silicon (see Fig. 1a and Supplementary Note 5). The spectral peak is blue shifted during the etching process owing to the increased transmission of silicon for short light wavelengths (see Fig. 1b)[56]. Our results reveal that the spectral peak $\lambda_p$ can be used to precisely measure the remaining thickness $H$ because of their close correlation as follows (see Fig. 1c and Supplementary Note 6):

$$\lambda_p = -54.0 e^{\frac{1270-H}{2090.67}} + 574.0 \qquad (1)$$

A local etching process was subsequently performed using an electro-optical system comprising a narrowband blue light source for generating a near-field diffraction pattern on the silicon/etchant interface and an electrical circuit for applying a forward bias $V_P$ of 800 mV and measuring the current signal $I_p$ (see Fig. 1a and Supplementary Note 5). When the silicon sample is illuminated, the photovoltage $V_o$ induced by the separation of the photogenerated hole-electron pairs partially offsets the applied potential $V_P$, resulting in a slower etching rate (i.e., photoinhibition or photovoltaic electrochemical etch-stop effect) on bright areas than on dark areas of the diffraction pattern around the pyramid tip[57,58]. The photoinhibition effect assisted KOH etching rate is found inversely related to the optical power (see Supplementary Note 7). In addition, our calculations indicate that the remaining thickness of silicon $H$ determines the size of the central dark area on the silicon/etchant interface, which in turn defines the fast-etching area and eventually the pore size of MPSNs (see Methods and Supplementary Note 8).

After the injection of KOH etchant and KCl electrolyte, the etching time required for perforation (i.e., the irradiation time) depends on the remaining thickness $H$ achieved during the pre-etching process. For example, a 120-s irradiation time is required for 114 nm of the remaining silicon thickness. Perforation leads to an over-etching stage characterised by rapid ramping in the current signal and an over-etching time that can be used to control the final pore size of MPSNs (see Fig. 1d). For instance, when using a silicon sample with a 19-nm remaining thickness, a 4.4-nm MPSN can be fabricated using an irradiation time of 20 s and an over-etching time of 0.6 s (see Fig. 1e–g).

## Characterisation of MPSNs

To validate the proposed fabrication method, MPSNs with pore sizes ranging from 4.4 to 64.2 nm were fabricated using different combinations of remaining thicknesses (or irradiation times) and over-etching times (see Fig. 2a and Supplementary Note 9). For instance, sub-10-nm pore sizes can be achieved using the same over-etching time of 0.6 s for different remaining thicknesses of silicon samples, such as 76 nm in $S_1$. To verify the photoinhibition effect, an over-etching time of 1 s was tested on the remaining silicon thickness of 114 nm in $S_2$, resulting in a nanopore with a central perforation and ring-shaped etched area. Noteworthily, the etched pattern in $S_2$ was consistent with the calculated near-field diffraction pattern, confirming that the photoinhibition effect regulated the etching rate and area (see Methods and Supplementary Note 8). Additionally, using an over-etching time longer than 1 s on a silicon sample with a 19-nm remaining

thickness could aid the fabricate of MPSNs with a greater than 40 nm pore size. The shapes and pore sizes of the example MPSNs $S_1$, $S_2$, and $S_3$ were further verified using electron transmission intensity profiles (i.e., grayscale images, see Fig. 2b–d and Supplementary Note 10).

A correlation study was conducted to examine the relationship among the pore size, remaining thickness and over-etching time. Our results reveal that the proposed fabrication method offers three distinct working regions mainly governed by the over-etching time. These regions include a sub-10 nm nanopore region using an over-etching time of ≤0.6 s, an over-40-nm nanopore region using an over-etching time of ≥2 s and a transition region using an over-etching time between 0.6 and 2 s (see Fig. 2e). Because the silicon sample has a defined remaining thickness, the final pore size can be precisely controlled by fine-tuning the over-etching time (see Fig. 2f). Additionally, a repeatability study assessing the proposed MPSN fabrication method demonstrated that the measured pore sizes with an average standard deviation of about 1.1 nm could be achieved by using four identical fabrication conditions, i.e., the same over-etching time of 0.6 s utilised to etch three samples of the same remaining thickness (see Fig. 2e, f and Supplementary Note 11).

## In-tube MPSN for inertial-kinetic translocation

To achieve controllable translocation of molecules, an in-tube MPSN device was developed to implement inertial kinetically regulated molecular translocation (i.e., inertial-kinetic translocation) in a laboratory-grade centrifuge (see Fig. 3a and Supplementary Note 12). While the conventional electrokinetic translocation behaviour of molecules is governed by electrophoretic and electroosmotic forces in nanopores[59], the inertial-kinetic translocation of molecules is mainly regulated by centrifugal force, with superposed counter-balancing forces caused by electrokinetic effects in the MPSNs (see Fig. 3a). This balanced state can be achieved by adjusting the pH value of the analyte medium and measuring the characteristic pH value for each molecule under test before the sensing experiment (see Supplementary Note 13)[60].

When an MPSN is centrifuged, its pyramid design offers a favourable funnel-shaped centrifugal force field that effectively captures and guides molecules through the nanopore by overcoming Brownian motion and a potential barrier $\Delta U$ caused by molecule–pore interactions (see Fig. 3a). Here, both the nanopore baseline current level $I_{base}$ and the nanopore root-mean-square current noise $I_{RMS}$ were proven to be independent of speed (see Supplementary Note 14). Additionally, the deep pyramidal structure of the MPSNs (approximately 21.25-μm deep) offers an elongated sensing length supporting longer dwell times. The calculation indicates an approximately 115-nm sensing length for an MPSN with a 15-nm pore size (see Methods and Supplementary Note 15). The measured current blockade signals clearly delineate three distinct stages related to the inertial-kinetic translocation in MPSNs: (i) molecule outside the sensing zone, (ii) molecule inside the sensing zone and (iii) molecule passing through the nanopore (see Fig. 3b). The molecular motions in stages (i) and (ii) are governed by centrifugal force and Brownian motion, whereas centrifugal force and the potential barrier become dominant in stage (iii). Compared with the inertial-kinetic translocation of molecules at the imbalanced states in MPSN, the frequency of molecular translocations is usually lower at the balanced state (see Fig. 3c–e). Such translocation frequency at the balanced state is also found depending on the molecular concentration $C_m$ and rotation speed (see Supplementary Note 16). In contrast to electrokinetic translocation in a conventional solid-state nanopore system[61,62], the inertial-kinetic translocation exhibits a capture radius that spans the entire upper flow chamber. This enables a significantly enhanced capture efficiency of single molecules, even at low concentrations of approximately 0.1 nM (see Supplementary Note 17). In addition, the inertial-kinetic

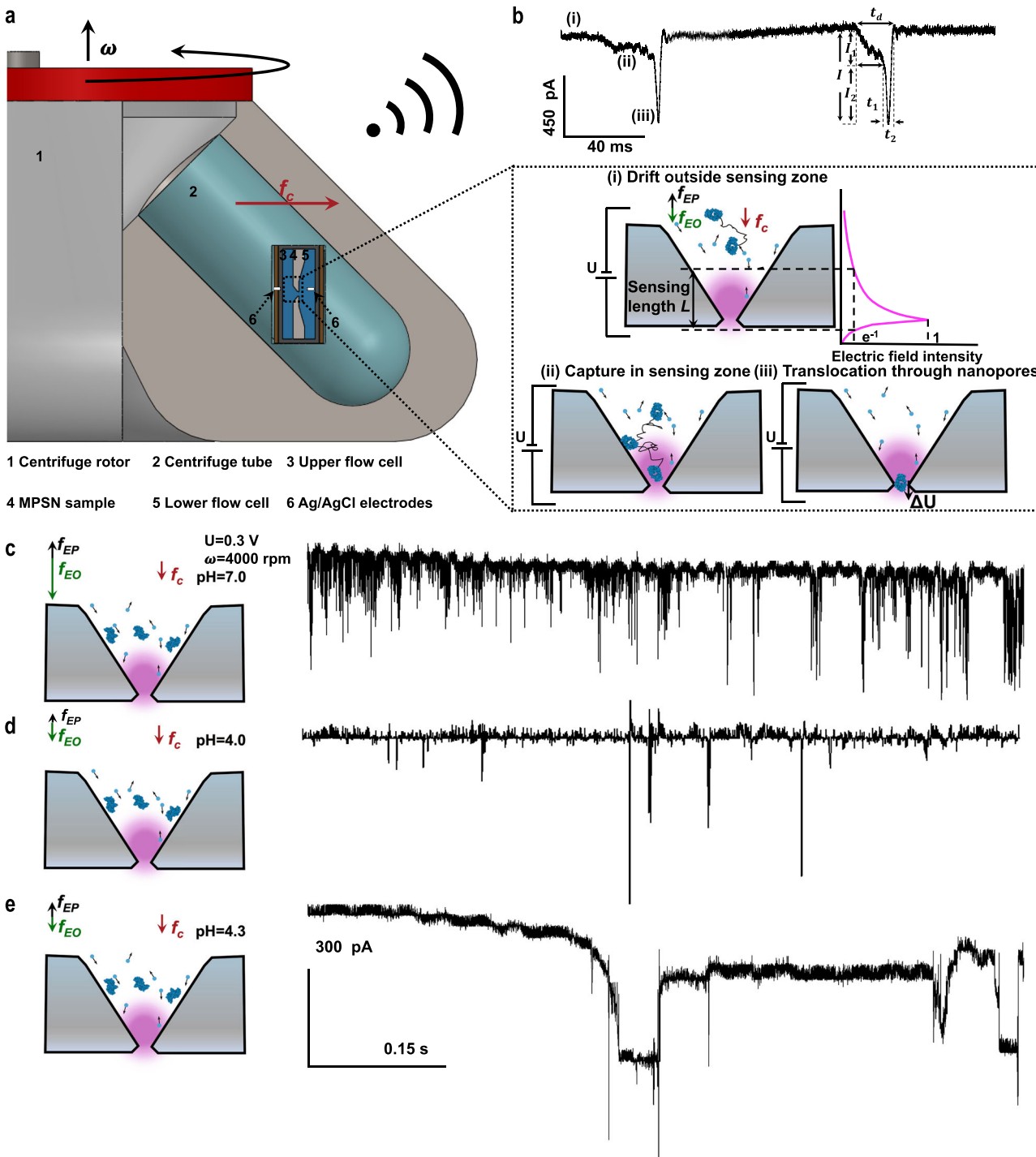

**Fig. 3 | In-tube molecular translocation and current blockade signals received by wireless transmission. a** Schematic representation of an in-tube MPSN device in a centrifuge. The device features a nanopore in a flow cell and two Ag/AgCl electrodes in KCl solution that apply voltage bias (U) and measure current blockade. The rotation speed is denoted by $\omega$ and the centrifugal force is defined by $f_c = m\rho\omega^2$. Here, $m$ is the molecular weight; $\rho$ is distance between the targets and the rotating shaft of the centrifuge, i.e., 25.6 cm in the experiments; and $\rho\omega^2$ at rotation speeds $\omega$ of 1000, 2000, 3000, and 4000 rpm are 280 g, 1120 g, 2520 g, and 4480 g, respectively. **b** Illustration of molecular motions in MPSN and associated current blockade signals received via wireless communication. The adjustment of pH value of the analyte medium leads to a counter-balanced electrophoretic force $f_{EP}$ and electroosmosis force $f_{EO}$, the centrifugal force $f_c$ dominates and competes

with Brownian motion to drive the outside molecules (i) into the sensing zone (ii) of a sensing length $L$, defined as the distance between two points where the maximal electric field intensity decays to its $e^{-1}$ value. Molecules are then translocated through the nanopore (iii) by overcoming the potential barrier $\Delta U$. Each molecular motion is reflected in the recorded current blockade signal, including the current baseline related to stage (i), the first current drop and duration ($I_1$, $t_1$) associated with molecular capture in stage (ii), and the second current drop and duration ($I_2$, $t_2$) due to molecular translocation in stage (iii). The amplitude of current blockade $I$ is defined as $I_1 + I_2$ and the dwell time $t_d$ is calculated as $t_1 + t_2$. **c–e** Schematics (left) illustrating competing forces extorted on BSA at different pH values and corresponding current traces (right) measured at rotation speed of 4000 rpm and voltage bias of 0.6 V. The balanced state of $f_{EP}$ and $f_{EO}$ is achieved at pH = 4.3 for BSA.

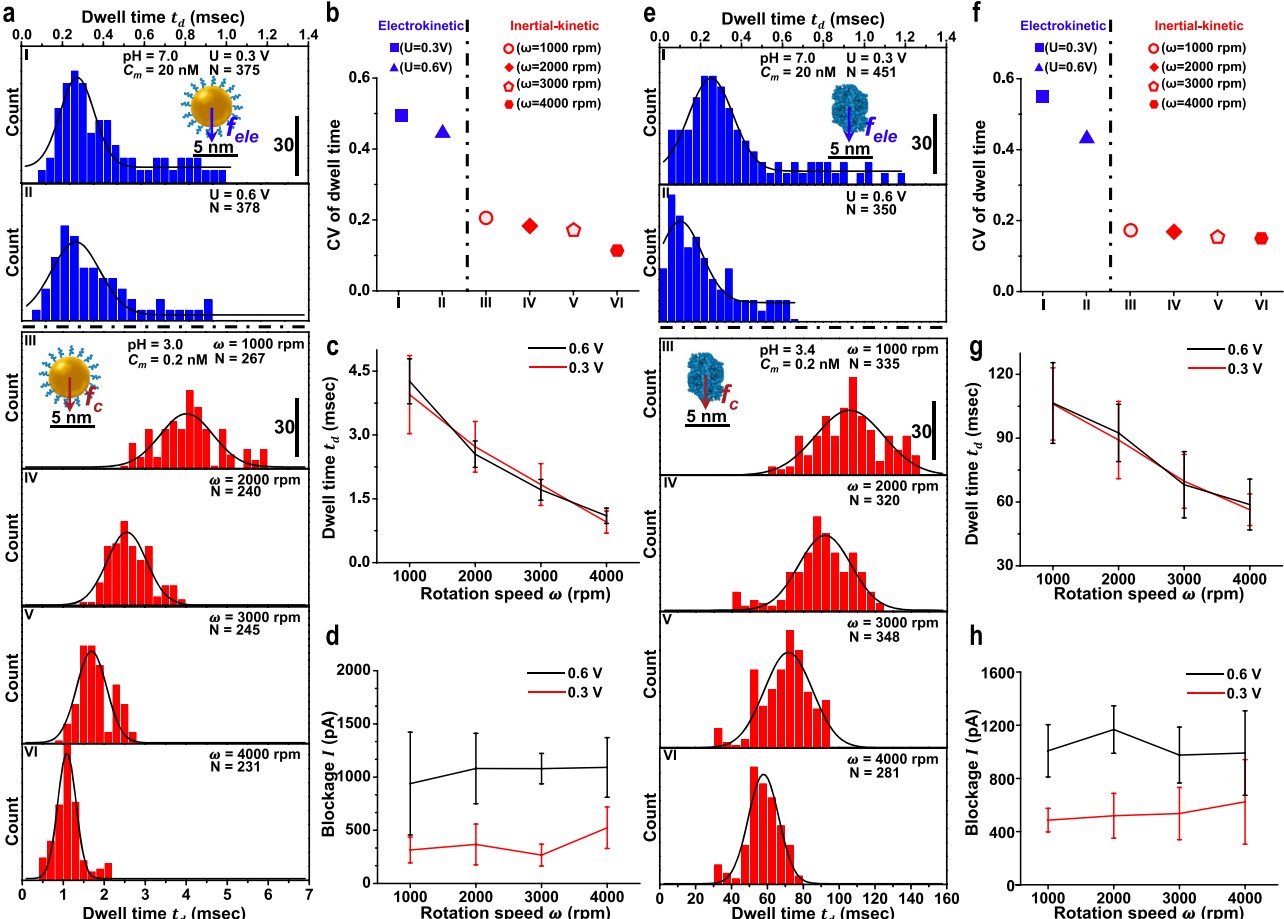

**Fig. 4 | Comparison of molecular translocation in MPSN using electrokinetic force and inertial-kinetic mechanism. a** Dwell time histograms of Au@PEG measured using electrokinetic translocation (blue) versus inertial-kinetic translocation (red) in a 15-nm MPSN with fitted Gaussian distribution curves. Electrokinetic translocation of Au@PEG with molecular concentration $C_m$ of 20 nM was performed under voltage biases U of 0.3 V (I) and 0.6 V (II) when pH is 7.0, while inertial-kinetic translocation of Au@PEG (III-VI) with $C_m$ of 0.2 nM was conducted under different rotation speeds $\omega$ with a pH of 3.0. **b** Comparing the variation of coefficient (CV) of dwell time in **a** to study the stability of electrokinetic translocation versus inertial-kinetic translocation of Au@PEG in MPSN. Dwell time $t_d$ (**c**) and blockage $I$ (**d**) of Au@PEG as a function of the rotation speed $\omega$ under voltage biases U of 0.3 V and 0.6 V. Panels **e** and **f** show the similar study as displayed in **a** and **b** to demonstrate the long and stable dwell times achieved using inertial-kinetic translocation of EpCAM in MPSN; with $C_m$ = 20 and 0.2 nM for electrokinetic and inertial translocation, respectively; N indicates events number in all histogram. Dwell time $t_d$ (**g**) and blockage $I$ (**h**) of EpCAM as a function of the rotation speed $\omega$ under voltage biases U of 0.3 V and 0.6 V. Error bars in **c**, **d**, **g**, **h** represent the standard deviations of experimental data measured from the blockade signals.

translocation at the balanced state offers a significantly longer dwell time than these obtained at the imbalanced states (see Fig. 3c–e).

Gold nanoparticles coated with polyethylene glycol (Au@PEG) and epithelial cell adhesion molecules (EpCAM) were tested using an in-tube device to compare the electrokinetic translocation with the inertial-kinetic translocation in MPSN (see Methods and Supplementary Note 16). The dwell time histograms of Au@PEG and EpCAM indicate that both the mean and standard deviation of dwell times inversely depend on the bias voltage in the electrokinetic translocation and the rotation speed in the inertial-kinetic translocation (see Fig. 4a, e). At a speed of 1000 rpm, the inertial-kinetic translocations of Au@PEG and EpCAM offer mean dwell times of approximately 4 ms and 100 ms, respectively. This corresponds to at least an order of magnitude improvement in dwell times than those can be achieved using the electrokinetic translocation at bias voltage of 0.3 V.

To assess the stability of single-molecule translocation, we used the coefficient of variation (CV) of dwell times (see Fig. 4b, f). The enhanced stability of the inertial-kinetic translocation offers CVs of dwell times that are at least 2 times lower than those measured using the electrokinetic translocation. The controllability of the inertial-kinetic translocation in terms of dwell times and amplitudes of current

blockades was further demonstrated by manipulating the rotation speed and bias voltage (see Fig. 4c, d, g, h). The dwell times were found almost independent of the bias voltages, whereas the amplitudes of current blockades were nearly irrelevant to rotation speeds in the inertial-kinetic translocation events. This offers an approach to tune and customise the desired current blockade signals to achieve an optimal conformational sensitivity for sensing molecules. Noteworthily, the dwell times of Au@PEG and EpCAM differ by more than one order of magnitude because they have different molecular weights despite their similar sizes of approximately 5 nm.

## Conformational sensing of macromolecules

An in-tube device made of a 15-nm MPSN was validated for its ability to sense six representative macromolecules of various masses and shapes, including EpCAM ($m$ = 37 kDa, $\beta$ = 0.72, $\xi$ = 0.0254 kDa$^{-1}$)[63], bovine serum albumin ($m$ = 67.5 kDa, $\beta$ = 0.57, $\xi$ = 0.0136 kDa$^{-1}$)[64], fragment antigen-binding ($m$ = 50 kDa, $\beta$ = 1.7, $\xi$ = 0.0231 kDa$^{-1}$)[1], streptavidin ($m$ = 60 kDa, $\beta$ = 0.9, $\xi$ = 0.0163 kDa$^{-1}$)[1], 4.8-nm spherical Au@PEGs ($m$ = 550 kDa, $\beta$ = 1, $\xi$ = 0.0018 kDa$^{-1}$)[65] and amylase ($m$ = 56 kDa, $\beta$ = 1.8, $\xi$ = 0.0209 kDa$^{-1}$)[1]. For each macromolecule, a low molecular concentration $C_m$ of approximately 0.1 nM was used to minimise the molecular interactions and was injected simultaneously into both

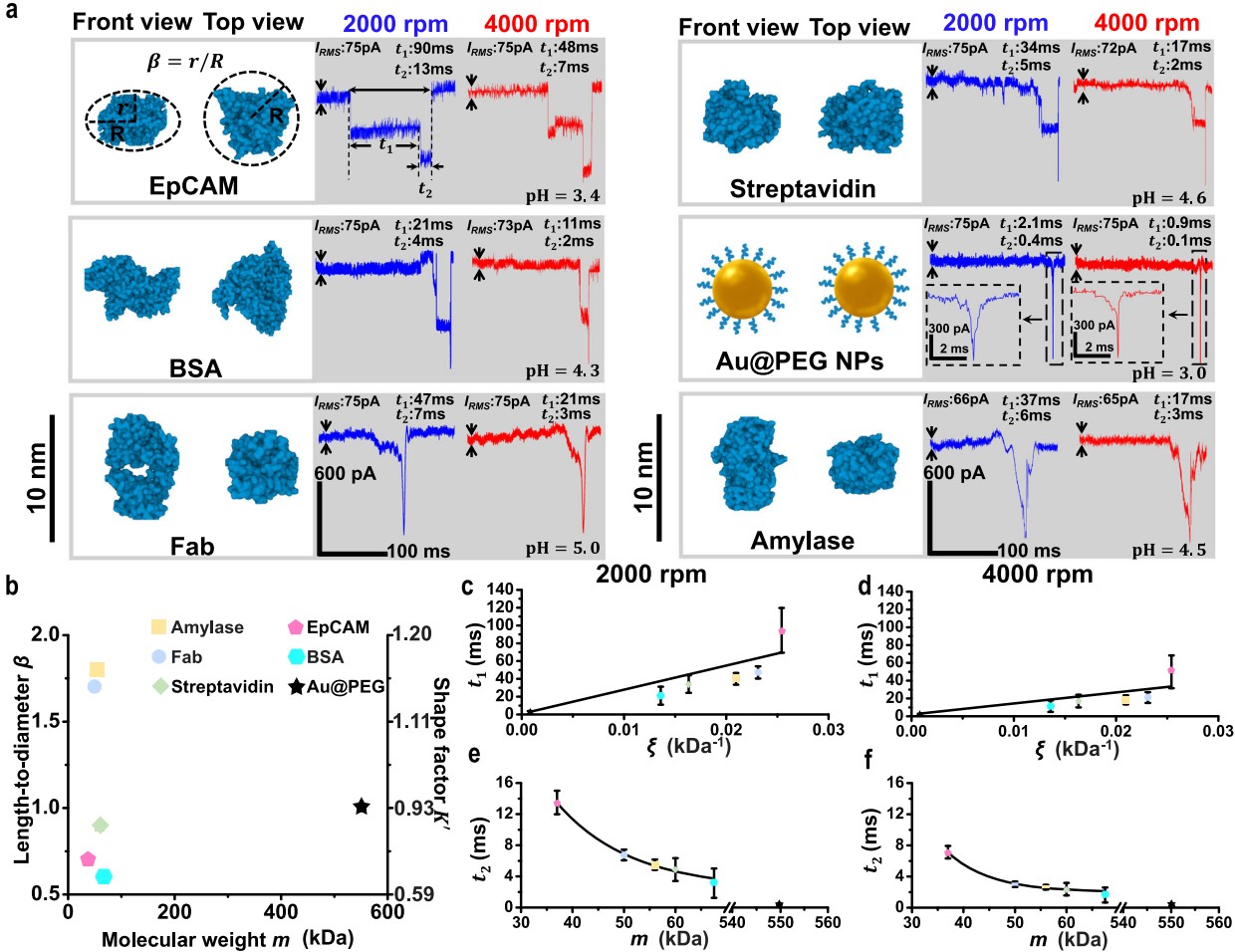

**Fig. 5 | Example molecular sensing using 15-nm MPSN. a** Current blockade signals (right) of six representative macromolecules measured at a voltage bias of 0.6 V and respective pH values of their balanced states in 1 M KCl solution under rotation speeds of 2000 rpm (blue) and 4000 rpm (red). Corresponding molecular/protein structures (left, from Protein Data Bank) are shown in front and top views. $\beta$ represents the length-to-diameter ratio of the molecule (i.e., $r/R$), where $r$ and $R$ are the polar and equatorial semi-axis of a molecule, respectively. $I_{RMS}$ indicates the root-mean-square current noise of each measurement. **b** Mass and shape distribution of tested molecules. A shape-to-mass weighting factors ($\xi$), i.e., $K'/m$ can be deifined for discriminating the six macromolecules. Here $K'$ is the shape factor determined solely by $\beta$, $m$ is molecular weight. Duration $t_1$ (**c**, **d**) and $t_2$ (**e**, **f**) measured at two rotation speeds, i.e., 2000 rpm (**c**, **e**) and 4000 rpm (**d**, **f**). Fitted curves (black) indicate that duration $t_1$ linearly increase with $\xi$ while duration $t_2$ decrease exponentially with $m$. Error bars in **c**–**f** represent the standard deviations of experimental data measured from the blockade signals.

the upper and lower flow cells of the in-tube device (see Fig. 3a). Molecular translocation was subsequently performed by exerting different centrifugal force $f_c$ at two rotation speeds (see Supplementary Note 18). The analysis of molecular motions in the nanopores indicated that the durations $t_1$ and $t_2$ are related to the molecular weight $m$, molecular shape factor $K'$, and rotation speed $\omega$ (see Methods and Supplementary Note 19). Theoretically, $t_1$ is linearly related to the molecular shape-to-mass weighting factors $\xi$ while $t_2$ exponentially dependences on the molecular weight $m$. The current blockade signals measured at a relatively low sampling rate of 50 kHz verified this, demonstrating unique trace features and distinct $t_1$ and $t_2$ of each molecule under test (see Fig. 5a). In contrast, when a low sampling rate was used, the electrokinetic translocation of macromolecules resulted in current blockade traces of low conformation-sensitive temporal features owing to fast molecular translocation in the nanopores (see Supplementary Note 3).

To verify the capability of the in-tube MPSN device for characterising the mass (i.e., $m$) and shape (i.e., $\xi$, see Fig. 5b) of macromolecules, the linear relationships between $t_1$ and $\xi$ and the negative exponential correlations between $t_2$ and $m$ were studied at different $\omega$ of 2000 rpm and 4000 rpm (see Fig. 5c–f). For example, for the tested

molecules, the fitted lines of $t_1$ versus $\xi$ were described as follows (see Fig. 5c, d):

$$t_1 = 2874.63\xi \ (\omega = 2000 \ rpm) \tag{2}$$

$$t_1 = 1319.42\xi \ (\omega = 4000 \ rpm) \tag{3}$$

These results indicate that $t_1$ linearly increases with the molecular shape-to-mass weighting factor and is adjustable using different rotation speed.

For the correlation between $t_2$ and $m$, the fitted curves can be described as follows (see Fig. 5e, f):

$$t_2 = 45.83e^{-\frac{m}{26.37}} \ (\omega = 2000 \ rpm) \tag{4}$$

$$t_2 = 419.20e^{-\frac{m}{8.39}} \ (\omega = 4000 \ rpm) \tag{5}$$

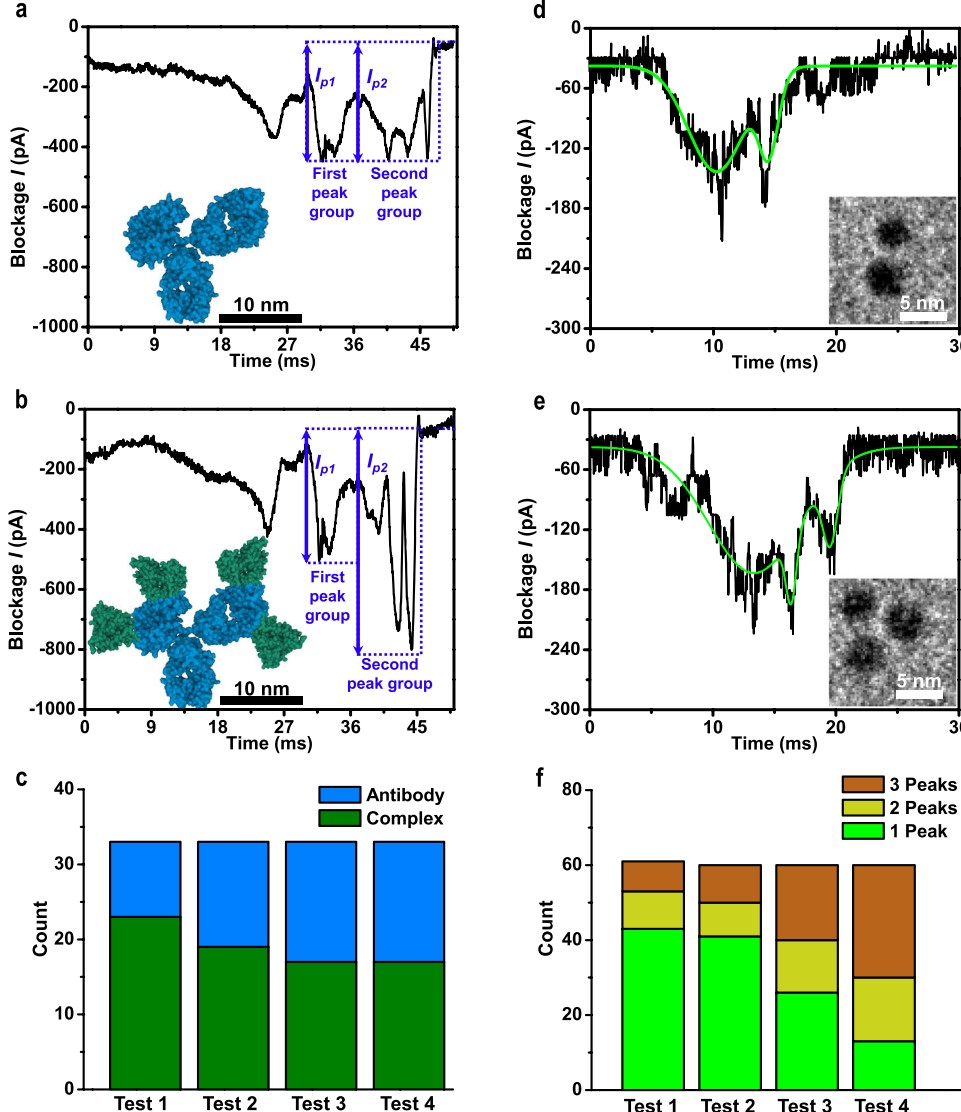

**Fig. 6 | Sensing molecular reactions and conformation changes of antibody-antigen complex and Au@PEG aggregates using 15-nm MPSN.** Current blockade signals for antibody EpCAM IgG (**a**) and antibody-antigen complex EpCAM IgG-EpCAM (**b**). $I_{p1}$ and $I_{p2}$ indicate the current amplitude of the first and second characteristic peak group, respectively. Insets show the corresponding molecular structures. **c**, Longitudinal monitoring of the dissociation of antibody-antigen complex with four tests performed in every 20 min. Current blockade traces of Au@PEG nanoparticles in bimolecules-aggregate (**d**) and trimolecules-aggregate (**e**), with corresponding molecular aggregation structures shown in insets. **f** Longitudinal sensing of aggregate of Au@PEG nanoparticles with four tests performed in every 10 min. All current blockade signals were measured at a voltage bias of 0.3 V and a rotation speed of 4000 rpm, while a pH value of 6.6 and 3.0 in 1 M KCl solutions were used for sensing antibody-antigen complex and Au@PEG nanoparticles, respectively.

These equations demonstrate that $t_2$ decays exponentially with the molecular weight and a fast decay rate can be achieved with a higher rotation speed.

### Longitudinal sensing of molecular reactions and conformation changes in molecular complexes

Molecular fingerprinting of the in-tube MPSN device can help longitudinally monitor conformational changes resulting from molecular reactions and interaction processes, such as the dissociation of EpCAM IgG (antibody) and EpCAM (antigen) complex. To prepare the antibody-antigen complex, EpCAM was incubated with 1 μM EpCAM IgG in 1 M KCl (pH = 7.4) for 45 min. The resulting reactant was diluted to 1 nM and injected into the flow cell of the in-tube device for sensing after adjusting the pH value to a balanced state by adding HCl (see Supplementary Note 20). By comparing the characteristic current blockade traces of the antibody molecule and antibody-antigen complex, it was found that the blockade signal of the complex exhibited a larger ratio of

two current amplitudes ($I_{p2} / I_{p1} \geq 1.5$) than of the antibody ($I_{p2}/I_{p1} \leq 1.0$) (see Fig. 6a, b). These features in the blockade traces provide an approach for evaluating the binding kinetics of the antibody-antigen complex. Moreover, the bimodal signal characteristics of IgG is calculated by using COMSOL Multiphysics (see Supplementary Note 21). In our study, the sensing experiment was repeated every 20 min and performed four times in total. It is well known that the EpCAM IgG-EpCAM complex gradually dissociates over time at the balanced state (pH = 6.6). This dissociation phenomenon was verified, and the ratio of the complex to antibody in the solution was observed to decrease gradually with time (Fig. 6c). The experiment was repeated in 20 nm and 23 nm MPSNs to test systematic errors caused by nanopore sizes (see Supplementary Note 22).

Furthermore, the in-tube MPSN device can be used for the longitudinal sensing of conformational changes in molecules during the aggregation and polymerisation processes. In this study, we investigated the aggregation of Au@PEG nanoparticles. The initial Au@PEG

solution was first subjected to ultrasonication, and the sensing experiment was repeated every 10 min and performed four times in total (see Supplementary Note 20). The current blockade signals of Au@PEG in single nanoparticle, biomolecular aggregates and trimolecular aggregates display features of a single-peak, two-peaks and three-peaks respectively (see Figs. 5a and 6d, e). The correspondence is proven by the blockade signals of the aggregations purified by centrifugation separation protocols (see Supplementary Note 23). Moreover, the signal characteristics corresponding to Au@PEG nanoparticles in trimolecules-aggregate is also calculated by using COMSOL Multiphysics (see Supplementary Note 21). By using these features in current blockade, the proportion of the bimolecular and trimolecular aggregates in the Au@PEG solution was found to gradually increase and eventually exceed that of single nanoparticle over time (Fig. 6f).

## Discussion

To achieve inertial-kinetic translocation of molecules, the nanopore needs to provide enough mechanical strength to withstand high levels of inertial force in centrifuge. In addition, a funnel-shaped nanopore is preferred to enable targets distributed over a large range for better capture and sense of molecules in nanopore. A solid-state funnel-shaped nanopore of over tens micrometres thickness can meet these requirements, and such a nanopore can be fabricated by using glass pulling and silicon chemical etching techniques[36,66]. However, conventional chemical etching techniques are not able to reliably fabricate silicon nanopores smaller than 8 nm[37–39]. Therefore, we propose photovoltaic electrochemical etch-stop technique, facilitating the controllable and reliable fabrication of funnel-shaped silicon nanopore of structure thickness up to 21.25 μm at resolution of about 4 nm. The core mechanism of this method involves the near-field diffraction-induced photoinhibition effect at the semiconductor/etchant interface, which defines the etching rate and area and ultimately regulates the pore size. Additionally, the micro-pyramid structure of MPSNs facilitates the capture of nano-sized molecules and their translocation through nanopores while offering a long sensing length.

We further propose a novel in-tube MPSN device for operating nanopore inside a centrifuge tube and implementing inertial-kinetic molecular translocation. This approach decouples molecular translocation from signal detection in the system and offers enhanced controllability over the translocation speed by conveniently regulating the rotation speed and centrifugal force in MPSN. While an electric field is still applied to MPSNs and used as a sensing method in our in-tube device, electrokinetic motion is effectively balanced by adjusting the pH values of the electrolyte used. Our experiments and theoretical studies confirm that the integration of centrifugation-governed inertial-kinetic translocation in MPSNs offers highly distinguishable molecular fingerprints related to the mass, shape, molecular reactions, and conformational change of proteins, with stable and adjustable dwell times and amplitudes of current blockade. This potentially allows for higher conformational sensitivity and temporal resolution than these offered by the conventional electrokinetic approaches to enable the accurate discrimination of proteins.

Despite demonstrating the feasibility of MPSNs and the in-tube device for the conformational discrimination of macromolecules, the current study has some limitations. To make it practical for use, the throughput of the device should be improved through parallel fabrication and by monitoring 2D nanopore arrays. Additionally, an intelligent data-driven close-loop strategy should be used to increase the controllability and sensing capability of our in-tube MPSN device. This will enable automated and quantitative recognition of molecules and facilitate the spatiotemporally resolved sensing of molecule interactions.

## Methods

### The preparation process in MPSN fabrication

Initially, two 200-nm-thick $Si_3N_4$ layers were deposited on both sides of a silicon sample using plasma-enhanced chemical vapour deposition (Oxford Instruments plc). Subsequently, two square patterns were transferred to the photoresist layers on both sides of the silicon through photolithography (SUSS MicroTec SE), which included standard procedures such as soft baking, UV exposure, developing and hard baking. The square pattern on the front side had a side length of 30 μm, whereas that on the backside had a side length of 800 μm. Carbon tetrafluoride (flow rate: 100 sccm; gas generation time: 20 min) and argon (flow rate: 5 sccm; gas generation time: 20 min) were then introduced into a plasma chamber (Plasma Etch Inc.) at a vacuum pressure of 100 mTorr to eliminate the exposed $Si_3N_4$ into a plasma environment excited by a radiofrequency electrostatic field (frequency: 3.0 MHz; output power: 29.6 W; radiofrequency generation time: 5 min). Finally, an inverted micro-pyramid structure was fabricated on the front side of the silicon sample by etching it with 30% (w/w) KOH in water heated to 80.0 °C on a hotplate.

### Molecule sample preparation

The proteins used in this study, including EpCAM, bovine serum albumin, streptavidin, EpCAM-antibody and amylase, were obtained from Thermo Fisher Scientific, Inc. Fragment antigen-binding was procured from Rockland Immunochemicals, Inc. The 4.8-nm spherical Au@PEGs were synthesised using a kinetically controlled seeded growth method. Initially, 3.5-nm-sized Au seeds were synthesised by injecting tetrachloroauric acid into a mixed solution of sodium citrate and tannic acid at 70 °C. Subsequently, the size of the Au seeds was increased to 4.8 nm by diluting the seed solution and injecting aliquots of gold precursor[67].

The molecule samples, comprising proteins and gold nanoparticles, were initially dispersed in 1 M KCl. The pH values of the samples were then adjusted by injecting HCl and precisely calibrated using a pH metre (PH-3C, Yk Scientific Instrument Co., Ltd) with a ± 0.01 pH resolution. The pH dependence of the Zeta potential of the tested samples was measured using a light scattering analyser (DelsaMax PRO, Beckman Coulter, Inc.). To determine the characteristic pH value of the balanced state between the electrophoretic and electroosmotic forces, nanopore sensing of the molecules at different pH values in MPSNs was performed by applying voltage only without the use of centrifugal forces. The characteristic pH value was identified as the corresponding value when the current blockade signal indicated minimal or no translocation events. This value was measured for each sample.

### In-tube nanopore sensing device

The in-tube device primarily comprises a centrifuge tube with a flow cell and an MPSN sample inside (see Fig. 3a). An electronic module was used to apply a voltage bias to the MPSNs and measure the current blockade signal through two Ag/AgCl electrodes. The measured blockade signals were first amplified at a sensitivity of −1.081 V/nA using an ultra-low bias preoperational amplifier (OPA128, Burr-Brown Corp.). The signal was then cascaded to a differential circuit with a low circuit noise of 0.4 pA (in root mean square at a sampling rate of 50 kHz). The amplified blockade signals are then digitised using an analogue-to-digital converter (AD9410, Analog Devices, Inc.) at a sampling rate of 50 kHz before being wirelessly transmitted to an external receiver in a mobile phone through an in-tube Bluetooth module (HM-BT4501, HOPE Microelectronics Co., Ltd.) at a baud rate of 115,200 bits/second.

### COMSOL simulation

The near-field diffraction pattern on the silicon/etchant interface was simulated using a two-dimensional axisymmetric model in COMSOL Multiphysics. Ring-shaped diffraction patterns with a dark centre area

around the pyramid tip were observed for different remaining silicon thicknesses ranging from 10 nm to 200 nm. The diameter of the dark centre area was found to decrease with decreasing remaining thickness.

The inertial-kinetic translocation process in MPSNs was simulated using COMSOL Multiphysics to understand the sensing region of the micro-pyramidal structure and the centrifugal force exerted on the molecules. In our COMSOL model, ion diffusion in the electrolyte was calculated using the Nernst-Planck equation in the transport of diluted species module. The distribution of the electric field in the nanopore structure was calculated using the Poisson equation in an electrostatics module by setting the applied voltages U to 0.3 V or 0.6 V, pore size to 15 nm and conductivity of 1 M KCl solution to 111,000 μS/cm. The sensing length of the nanopore was defined as the length between two points at which the maximal electric field intensity decays to its $e^{-1}$ value[40,68]. Additionally, the distribution of funnel-shaped centrifugal force field and drag force generated on the molecules in the micro-pyramidal structure were calculated using Navier-Stokes equation in the laminar flow module.

### Modelling of molecular motions in MPSNs

During the molecular capturing process (see Fig. 3b-ii), molecular motions are governed by a competition between centrifugal force and Brownian diffusion and can thus be described using the Langevin equation. Moreover, based on the Einstein–Smoluchowski equation, the molecular diffusion coefficient is closely related to the molecular shape $K'$. Therefore, the capture duration $t_1$ can be calculated using the equation $6\pi\mu RK'L/f_c$, where $\mu$ represents dynamic viscosity coefficient of molecules, $R$ represents the equatorial semi-axis of molecules, $L$ represents the sensing length and $f_c$ represents the centrifugal force exerted on molecules. As for molecular motions during the translocation through nanopore process (see Fig. 3b-iii), translocation time $t_2$ (time until desorption) is a stochastic variable dependent on the bulk dissociation rate.

Thus, $t_2$ can be calculated using an Eyring-like form $t_0 e^{-\frac{h(f_c)}{kT}}$, where $h$ represents the force-dependent factor, $t_0$ represents the mean of the exponential distribution, $k$ represents Boltzmann constant and $T$ represents the environmental temperature. Consequently, the capture duration $t_1$ and translocation time $t_2$ can be calculated using the following equations:

$$t_1 = \frac{6\pi\mu R\xi L}{\rho\omega^2} \tag{6}$$

$$t_2 = t_0 e^{-\frac{h(m\rho\omega^2)}{kT}} \tag{7}$$

Here $\xi$ is the shape-to-mass weighting factors. The fitting results of Eqs. (2)–(5) lead to a sensing length $L$ in MPSN of about 92 nm, which is close to the calculated value of 115 nm, and a force-dependent factor $h$ of about 8.45 μm.

## Data availability

The main data generated in this study are available within the paper and the Supplementary Information. Source data are provided with this paper. Additional data related to this paper may be requested from the corresponding authors (W.Y., H.-P.H). Source data are provided with this paper.

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

## Acknowledgements

We thank Dr. Chung Hang Jonathan Choi and Dr. Cecilia Ka Wing Chan for providing the Au NPs for initial experiments; Dr. Yi-Ping Ho for obtaining Zeta-potential measurements. This work was supported in part by the Research Grant Council (RGC) of Hong Kong SAR through ECS project [24211020 (W.Y.)] and GRF projects [14207218 (H.-P.H.), 14207419 (H.-P.H.), 14204621 (H.-P.H.), 14203821 (W.Y.), 14216222 (W.Y.)], and the Innovation and Technology Commission (ITC) of Hong Kong SAR through ITF projects [ITS/137/20 (H.-P.H.), ITS/240/21 (W.Y.)], the Science, Technology and Innovation Commission (STIC) of Shenzhen Municipality through Shenzhen-Hong Kong-Macau Science and Technology Program (Category C) project [SGDX20220530111005039 (W.Y.)], the Brain Pool Fellowship program funded by the National Research Foundation of the Korean Government [2021H1D3A2A01099337 (H.-P.H.)].

## Author contributions

H.-P.H. and W.Y. conceived the idea of the work. J.Y., and T.P. fabricated the nanopore for the initial studies. W.Y. and H.-P.H. optimised experimental designs and fabrication protocols. J.Y. and T.P. investigated the in-tube device assembly. W.Y., and H.-P.H. discussed and improved the research mechanism. J.Y. and Z.X. performed nanopore test, signal characterisation, and other related experiments. J.Y. and T.P. prepared the manuscript with thorough editing and polishing from W.Y. and H.-P.H.

## Competing interests

The authors declare no competing interests.
