## [Peer Review File · Nature Communications]

REVIEWER COMMENTS

Reviewer #1 (Remarks to the Author):

The authors have addressed all my previous comments. I recommend publication on Nature Communications.

My remaining suggestion concerns the semantics. Technically, small molecules are defined as species with a low molecular weight (<1 kDa), while larger molecules like proteins are referred to as macromolecules. Therefore, it may be more appropriate for the authors to use "macromolecules" when referring to the analytes in general, or specify "proteins" as they are the primary test subjects.

Reviewer #2 (Remarks to the Author):

In this paper, Yang et al. developed fabrication method of crystal silicon nanopore using photoinhibition effect and demonstrate protein and Au nanoparticle detection using inertial force. In manuscript, nanopore fabrication with different etching time was demonstrated and pore dimension was characterized using Transmission electron microscope (TEM) image. Then, authors set fabricated pore in the centrifuge to apply inertial-kinetic force to the molecules to run through a pore. Their experimental results show that the detection method based on inertial kinetic force provides a blockade current, potentially relating to the mass, shape, and reaction kinetics of the protein. Although proposed methods are quite original and their results contribute to the field development and I am pretty sure that authors made tremendous improvement from previous manuscript submitted to Nature nanotechnology, I can still find missing experiments, misleading/missing references, and incorrect explanations in the current manuscript. Therefore, I would request to run more experiments to support the validity of their method and improve manuscript for publication in Nature communication.

My comments are the following;

Major Comments:

1. In Page 2 Line 28 authors said their fabricated pores shows excellent structural strength to withstand high inertial force. To show the stability of inertial force, it is better to try typical silicon nitride pore which has nanometer thickness for comparison. Also, references in SI Note are wrong, please correct all of them. For examples,

a. Ref.1-13 in SI did not use FIB.

b. Ref.11 in SI did not use controlled dielectric breakdown.

c. Ref. 18 is review paper. It is not research paper, please correct this.

d. There is no references in Photovoltaic etching part. Please add references there.

2. In Page 4 Line 8, what is the heating effect of laser on the silicon etching? Laser power is really high, so laser light should cause heating which can influence etching.

3. Authors measure pore size by using TEM. However, by my eye TEM image is too blurred. Authors should get highly contrasted TEM image having clear boundary of hole as people in field commonly obtained as you can find this paper (Park SR, Peng H, Ling XS (2007) Fabrication of nanopores in silicon chips using feedback chemical etching. Small 3:116–119). Maybe dispersion of nanoparticles would help for TEM beam alignment to get clear image.

4. How do authors obtain this equation(1)? There should be dispersion in the wafer thickness, etching orientation and SiN window size and so on, so I feel it is hard to say single nanometer levels resolution. Please check several wafer and tell the accuracy of this fitting profile.

5. In addition, authors determine pore size and thickness based on optical image and TEM image. However, since TEM and optical image are not clear, it is hard to say these images can tell accurate number of pore size and thickness. To support, pore dimension, authors should add voltage vs. current data for all pore conditions. Ionic current can tell much clear pore dimension because ion current is much sensitive to pore dimension.

6. In Page 6 Line 12, authors said pore size with a standard deviation of less than 1 nm, but I could not find such resolution based on Figure 2e and f. To propose such high fabrication resolution, authors should try same fabrication condition as least 3-4 times and show the accuracy (distribution) of this pore fabrication method.

7. In page 11 line 1, it is interesting to see that proposed system can detect molecular and particles at low concentration (0.1nM) with quite high frequency(capture rate). Authors should mention a reason

why this type of pore can detect such low concentration compared with typical solid-state nanopore system which has 1-25 nm pore with 1-50 nm thickness. Maybe COMSOL simulation might help to find capture radius of this type of pore.

8. In equation (2)-(5), authors show fitting result of t_1 and t_2 at 2000 and 4000rpm. My understanding is basically this sensing method involve inertial kinetic force because authors did not find any difference between 300 and 600 mV in Figure 4. Then why can different fitting equation be applied for t_1 and t_2 ?

9. Do experiments in Figure 6a and b use same pore? Ration different in antibody and antibody complex can be caused by pore dimension. If not used different pore, please run experiments and show the ratio at several different pores.

10. In page 13 line 20, how authors know single, two and three peaks single, dimer and trimer nanoparticles? To make sure of this assumption, authors should prepare purified dimer or trimer nanoparticle and run the control experiment. At least, dimer nanoparticles can be prepared. See this paper: K. Esashika, R. Ishii, S. Tokihiro, and T. Saiki, "Simple and rapid method for homogeneous dimer formation of gold nanoparticles in a bulk suspension based on van der Waals interactions between alkyl chains," *Opt. Mater. Express* 9, 1667-1677 (2019).

11. Also there is missing explanation that why EpCAM IgG and Au@PEG nanoparticles in trimolecules-aggregate showed two peaks and three peaks respectively. Authors need to refer previous works or explains detailed mechanism step by step.

12. In page 13 line 28, authors proposed silicon pore with microthickness provide high mechanical strength to withstand inertial force. To propose the high mechanical strength, authors should show pore expansion over the time by showing current trace at certain voltage and inertial force (rpm) for several time sections such as 0, 30, 60, 120 min.

13. In Figure 5, noise level for each trace is so different. Why such huge different happen? Difference in noise level for each experiment might be influenced in analysis. Please indicate distribution of noise level at each experiments.

Minor Comment:

1. Ref. 4 in Page1, Line 24 is wrong. It should be ref. 5.

2. In Page 1 Line 35-36, it is better to say “controlled molecular translocation”, not “controlled translocation” .

3. In Page 1 line 36, why authors only describe this paper? so far, there are so many approaches to slow down translocation speed. Here is examples;

a. Keyser UF et al (2006) Direct force measurements on DNA in a solid-state nanopore. *Nat Phys.* 2(7):473

b. Peng HB, Ling XS (2009) Reverse DNA translocation through a solid-state nanopore by magnetic tweezers. *Nanotechnology.* 20(18):185101

c. Nelson EM, Li H, Timp G (2014) Direct, concurrent measurements of the forces and currents affecting DNA in a nanopore with comparable topography. *ACS Nano.* 8(6):5484–5493

d. Hyun C et al (2013) Threading immobilized DNA molecules through a solid-state nanopore at >100 μ s per base rate. *ACS Nano.* 7(7):5892–5900

e. Akahori, R., Yanagi, I., Goto, Y. et al. Discrimination of three types of homopolymers in single-stranded DNA with solid-state nanopores through external control of the DNA motion. *Sci Rep* 7, 9073 (2017). <https://doi.org/10.1038/s41598-017-08290-6>

4. There is no axis descriptions on graph in SI note 6. Please add it otherwise I cannot understand what it is saying.

5. In page 8 line 14, this sentence is incorrect. I agree with elongated sensing length provide longer dwell time. However, the elongated sensing length does not provide better temporal resolution. If authors want to propose this concept, they should add additional explanation for it.

6. In Figure 4, authors should add events number in all histogram to show better statistically understanding. Also please add concentration of Au@PEG and EpCAM.

Reviewer #3 (Remarks to the Author):

The author carefully addressed the comments from the reviewers. There were issues with the references and claims especially in the introduction part, and the author have addressed most of them. The manuscript is much better now and can be published. I still recommend that the authors show the

detection of DNA using their setup, which is an essential work to demonstrate the capability of the method. This could be done in future work. Some minor comments are listed below before the publication.

1) Table S1. Please cite the reference directly and avoid using the indirect citation in Table S1 (ref 18), with regards to the "Glass pulling" and "Chemical etching". Where are the references for glass nanopores? The "Glass pulling" reference should also be cited in Page 13, Line 33: "fabricated by using glass pulling and silicon chemical etching techniques". References are missing here.

2) Page 13, Line 35. "conventional glass pulling techniques still need further optimization to consistently produce nanopores of same size and conventional chemical etching techniques are not able to reliably fabricate silicon nanopores smaller than 8 nm"

The authors still have dangerous claims without checking enough references on nanopore research.

First, Ulrich F. Keyser's group has shown that the pore size is pretty consistent with their typical pore size of 14 ± 3 nm.

Second, "chemical etching techniques are not able to reliably fabricate silicon nanopores smaller than 8 nm". Please add references here.

Please also thoroughly check other references in the manuscript.

Responses to the reviewers' comments

We are grateful to the reviewers' comments and recommendation for publication. We have revised the manuscript accordingly and the added or changed parts are marked in **red** in the manuscript text file. We have also provided the detailed responses as appended below. Author responses are in **blue**, changes and additions to the manuscript are in **red**, original texts in manuscript and reviewer's comments are in **black**, and all line numbers refer to the documents with redlined highlights.

Reviewer #1 (Remarks to the Author):

Comment 1:

The authors have addressed all my previous comments. I recommend publication on Nature Communications.

My remaining suggestion concerns the semantics. Technically, small molecules are defined as species with a low molecular weight (<1 kDa), while larger molecules like proteins are referred to as macromolecules. Therefore, it may be more appropriate for the authors to use "macromolecules" when referring to the analytes in general, or specify "proteins" as they are the primary test subjects.

Response:

We sincerely appreciate the reviewer for providing valuable feedback and recommendations for publication.

Based on the suggestion, we have revised the manuscript title to "In-tube micro-pyramidal silicon nanopore for inertial-kinetic sensing of single molecules." This change aligns with the reviewer's recommendation and enhances the clarity and specificity of the title. Additionally, we have carefully addressed the concern regarding the term "small molecules" in the revised manuscript. We have replaced this term with more appropriate and context-specific terms such as "macromolecules," "proteins," or "biomolecules" as applicable. This modification ensures accuracy and precision in describing the molecules under investigation.

We sincerely appreciate the reviewer's insightful comments, which have contributed to the overall improvement of the manuscript.

Reviewer #2 (Remarks to the Author):

In this paper, Yang et al. developed fabrication method of crystal silicon nanopore using photoinhibition effect and demonstrate protein and Au nanoparticle detection using inertial force. In manuscript, nanopore fabrication with different etching time was demonstrated and pore dimension was characterized using Transmission electron microscope (TEM) image. Then, authors set fabricated pore in the centrifuge to apply inertial-kinetic force to the molecules to run through a pore. Their experimental results show that the detection method based on inertial kinetic force provides a blockade current, potentially relating to the mass, shape, and reaction kinetics of the protein. Although proposed methods are quite original and their results contribute to the field development and I am pretty sure that authors made tremendous improvement from previous manuscript submitted to Nature nanotechnology, I can still find missing experiments, misleading/missing references, and incorrect explanations in the current manuscript. Therefore, I would request to run more experiments to support the validity of their method and improve manuscript for publication in Nature communication.

Response:

We would like to extend our sincere appreciation to the reviewer for the valuable and constructive feedback, which has significantly contributed to the improvement of our manuscript for publication. We have carefully reviewed and addressed all the concerns raised, making necessary revisions to references and explanations accordingly.

Furthermore, in order to strengthen the validity of our method, we have conducted additional experiments as suggested by the reviewer. These experiments have been carefully designed and executed to provide robust evidence supporting our findings.

We are confident that the revisions made, along with the additional experiments, have effectively addressed all the concerns raised by the reviewer. We are grateful for the thorough evaluation and insightful comments, which have undoubtedly enhanced the quality and rigor of our manuscript.

Once again, we express our sincere gratitude to the reviewer for his/her valuable contribution to our work.

Comment 1:

In Page 2 Line 28 authors said their fabricated pores shows excellent structural strength to withstand high inertial force. To show the stability of inertial force, it is better to try typical silicon nitride pore which has nanometer thickness for comparison.

Response:

We appreciate the reviewer for their excellent and constructive comment.

In response, we have included a statement regarding the structural strength of a typical silicon nitride pore for comparison in Supplementary Note 2. The added information can be found below:

“

Supplementary Note 2. Calculations of the mechanical strength of MPSN and a typical silicon nitride pore.

When the nanopore structure is subjected to the centrifugal force f_c exceeding the breaking load f_{load} ($f_c > f_{load}$), the nanopore will be broken^{34,35}. The relationship between the breaking load f_{load} and the ultimate compressive strength δ can be described as^{34,36}:

$$f_{load} = \frac{2w\delta T^2}{3L} \quad (S1)$$

where the ultimate compressive strength δ is decided by the material property, i.e., 6 GPa for single crystal silicon³⁴ and 3.4 GPa for Si₃N₄ deposited with PECVD³⁶; L is the structural length, such as 30 μm for the MPSN and 70 μm for a typical Si₃N₄ nanopore³⁷; w is the structural width, for example, 30 μm for MPSN and 70 μm for a typical Si₃N₄ nanopore³⁷; T is the structural thickness, which is 21.25 μm for MPSN and 30 nm for the Si₃N₄ nanopore³⁷.

According to Equation S1, the breaking load f_{load} of a typical Si₃N₄ nanopore ($w = 70 \mu\text{m}$; $L = 70 \mu\text{m}$; $T = 30 \text{ nm}$) and MPSN ($w = 30 \mu\text{m}$; $L = 30 \mu\text{m}$; $T = 21.25 \mu\text{m}$) can be calculated as 2.1 μN and 1.8 N, respectively.

Moreover, the centrifugal force f_c exerted on the nanopore samples can be described as:

$$f_c = m\rho\omega^2 \quad (S2)$$

where m is the molecular weight ($4.7 \times 10^{-13} \text{ kg}$ for the Si₃N₄ nanopore; $3.0 \times 10^{-11} \text{ kg}$ for the MPSN); ρ is the nominal rotation radius of the nanopore sample and can be assumed as the distance between the nanopore to the spinning center of centrifuge, i.e., about 25.6 cm in the experiments; ω is the rotational speed.

According to Equation S1 and S2, the Si₃N₄ nanopore and MPSN are able to maintain the mechanical stability, i.e., $f_c < f_{load}$, at rotation speeds ω below 3.6×10^4 and 3.8×10^6 rpm, respectively.

”

Newly added References 36, 37 related to a typical Si₃N₄ nanopore:

“

36 Gaspar, J., Paul, O., Chu, V. & Conde, J. Mechanical properties of thin silicon films deposited at low temperatures by PECVD. *Journal of Micromechanics and Microengineering* **20**, 035022 (2010).

37 Park, S., Lim, J., Pak, Y. E., Moon, S. & Song, Y.-K. A solid state nanopore device for investigating the magnetic properties of magnetic nanoparticles. *Sensors* **13**, 6900-6909 (2013).

”

Comment 2:

Also, references in SI Note are wrong, please correct all of them. For examples,

a. Ref.1-13 in SI did not use FIB.

b. Ref.11 in SI did not use controlled dielectric breakdown.

c. Ref. 18 is review paper. It is not research paper, please correct this.

d. There is no references in Photovoltaic etching part. Please add references there.

Response:

We appreciate the reviewer for bringing our mistakes to our attention. We have promptly rectified these errors in table S1. Thank you for your diligence in identifying these issues.

Table S1 has been updated as below:

“

Fabrication Methods	Material	Thickness	Pore size	Pore shape	Scalability/repeatability
FIB	SiN ¹⁻⁵	< 30 nm	[0.3 nm, 280nm] ^{1,2,6-8}	Cylindrical	Yes
	SiO ₂ ^{8,9}	[0.5 nm, 60 nm]		Cylindrical	
	SiC ^{6,10}	< 20 nm ⁶		Cylindrical	
	2D materials ^{7,11,12}	around 0.3nm ¹³		Cylindrical	
TEM shrinking	SiN	Not decided by the technique	> 0.13nm ¹⁴⁻¹⁶	Cylindrical	Yes
	Si			Cylindrical	
	SiO ₂			Cylindrical	
E-beam lithography	Si	50-100 nm ^{17,18}	5 -113 nm	Funnel-shaped	Yes
Controlled dielectric breakdown	SiN	>10nm ¹⁹	> 1.1 nm ¹⁹⁻²³	Cylindrical	Yes
	SiO ₂	>30nm ²⁰		Cylindrical	
	HfO ₂	>10nm ²¹		Cylindrical	
Glass pulling	glass ²⁴⁻²⁸	nanopore with over tens micrometers ²⁷	>11 nm ²⁸	Funnel-shaped	Yes
Chemical etching	Si ²⁹⁻³¹	nanopore with micro thick membrane ³⁰	> 8 nm ³¹	Funnel-shaped	Yes
Photovoltaic electrochemical etch-stop technique (our method) ^{32,33}	Si	over tens micrometers	> 4.5 nm	Funnel-shaped	Yes

”

Newly added References 1-13 related to FIB:

“

- 1 Fürjes, P. Controlled focused ion beam milling of composite solid state nanopore arrays for molecule sensing. *Micromachines* **10**, 774 (2019).
 - 2 Lo, C. J., Aref, T. & Bezryadin, A. Fabrication of symmetric sub-5 nm nanopores using focused ion and electron beams. *Nanotechnology* **17**, 3264 (2006).
 - 3 Gadgil, V., Tong, H., Cesa, Y. & Bennink, M. L. Fabrication of nano structures in thin membranes with focused ion beam technology. *Surface and coatings technology* **203**, 2436-2441 (2009).
 - 4 Sawafta, F., Carlsen, A. T. & Hall, A. R. Membrane thickness dependence of nanopore formation with a focused helium ion beam. *Sensors* **14**, 8150-8161 (2014).
 - 5 Rudenko, M., Yin, D., Holmes, M., Hawkins, A. & Schmidt, H. in *Ultrasensitive and Single-Molecule Detection Technologies II*. 130-140 (SPIE).
 - 6 Biance, A.-L. *et al.* Focused ion beam sculpted membranes for nanoscience tooling. *Microelectronic engineering* **83**, 1474-1477 (2006).
 - 7 Ibrahim, N. N. N. M. & Hashim, A. M. High sensitivity of deoxyribonucleic acid detection via graphene nanohole/silicon micro-nanopore structure fabricated by focused ion beam. *Materials Letters* **305**, 130740 (2021).
 - 8 Komarov, F. *et al.* Ion-beam formation of nanopores and nanoclusters in SiO₂. *Vacuum* **78**, 361-366 (2005).
 - 9 Fu, Y., Bryan, N. K. A. & Fatt, L. T. Fabrication and characterization of nanopore array. *Journal of nanoscience and nanotechnology* **6**, 1954-1960 (2006).
 - 10 Gierak, J. *et al.* Sub-5 nm FIB direct patterning of nanodevices. *Microelectronic engineering* **84**, 779-783 (2007).
 - 11 Morin, A. *et al.* FIB carving of nanopores into suspended graphene films. *Microelectronic Engineering* **97**, 311-316 (2012).
 - 12 Md Ibrahim, N. N. N. & Hashim, A. M. Fabrication of Si Micropore and Graphene Nanohole Structures by Focused Ion Beam. *Sensors* **20**, 1572 (2020).
 - 13 Waduge, P., Larkin, J., Upmanyu, M., Kar, S. & Wanunu, M. Programmed synthesis of freestanding graphene nanomembrane arrays. *Biophysical Journal* **108**, 330a (2015).
- ”

Newly added References 19-21 related to the controlled dielectric breakdown:

“

- 19 Kwok, H., Briggs, K. & Tabard-Cossa, V. Nanopore fabrication by controlled dielectric breakdown. *PLoS one* **9**, e92880 (2014).
- 20 Wang, Y., Chen, Q., Deng, T. & Liu, Z. Self-aligned nanopore formed on a SiO₂ pyramidal membrane by a multipulse dielectric breakdown method. *The Journal of Physical Chemistry C* **122**, 11516-11523 (2018).
- 21 Wang, Y., Chen, Q., Deng, T. & Liu, Z. Nanopore fabricated in pyramidal HfO₂ film by dielectric breakdown method. *Applied Physics Letters* **111** (2017).

”

Newly added Reference 32, 33 related to the Photovoltaic electrochemical etch-stop technique:

“

32 Strandman C & Backlund Y. Bulk silicon holding structures for mounting of optical fibers in V-grooves. *Journal of microelectromechanical systems* 6, 35-40 (1997).

33 Voss R, Siedel H & Baumgartel H. Light-controlled, electrochemical, anisotropic etching of silicon. In: *TRANSDUCERS'91: 1991 International Conference on Solid-State Sensors and Actuators. Digest of Technical Papers*. IEEE (1991).

”

Comment 3:

In Page 4 Line 8, what is the heating effect of laser on the silicon etching? Laser power is really high, so laser light should cause heating which can influence etching.

Response:

Thanks to the reviewer for the constructive comment.

Accordingly, we have calculated the temperature distribution on the silicon/etchant surface in the pre-etching process based on the Helmholtz equation and the heat conduction equation by using COMSOL Multiphysics^{2-1, 2-2}. The heat source is the absorption of incident light by the silicon layer and KOH. In the simulation, 460-nm bandwidth blue light (collimated beam size: 8.1 mm; output power: 0.77 W; light power density: 3.7 mW/mm²) is set as the far field and unfocused light source. The initial temperatures of both silicon sample and KOH solution are set to 293.15 K (20°C). The negative imaginary component of silicon's and 1M-KOH's refractive index (responsible for the light absorption) are about 0.077 and 0.02 at 460 nm^{2-3, 2-4}, respectively. The remaining layer thickness of silicon is set to be 1.0 μm.

Figure R2-1. Calculated temperature distribution around silicon nanopore by laser illumination in the pre-etching process.

As shown in Fig R2-1, the maximum temperature in the silicon sample is close to 295.03 K (21.9°C), in which the etch rate of 30% KOH for (1-0-0) silicon is 45-50 nm/min. Our further calculation indicates that the silicon layer temperature would be larger than 35°C only when

the optical power density is larger than 80 mW/mm^2 , which can be considered to significantly affect the etching rate of KOH for (1-0-0) silicon.

In addition, we have revised the Figure S3 to indicate the temperature during the etching process, as shown below:

“

”

Reference:

2-1 Baffou, G., Quidant, R. & García de Abajo, F. J. Nanoscale control of optical heating in complex plasmonic systems. *ACS nano* **4**, 709-716 (2010).

2-2 Zheng, J., Xing, X., Yang, J., Shi, K. & He, S. Hybrid optofluidics and three-dimensional manipulation based on hybrid photothermal waveguides. *NPG Asia Materials* **10**, 340-351 (2018).

2-3 Green, M. A. & Keevers, M. J. Optical properties of intrinsic silicon at 300 K. *Progress in Photovoltaics: Research and applications* **3**, 189-192 (1995).

2-4 Prusi, A., Arsov, L., Haran, B. & Popov, B. N. Anodic behavior of Ti in KOH solutions: ellipsometric and micro-Raman spectroscopy studies. *Journal of the Electrochemical Society* **149**, B491 (2002).

Comment 4:

Authors measure pore size by using TEM. However, by my eye TEM image is too blurred. Authors should get highly contrasted TEM image having clear boundary of hole as people in field commonly obtained as you can find this paper (Park SR, Peng H, Ling XS (2007)

Fabrication of nanopores in silicon chips using feedback chemical etching. Small 3:116–119). Maybe dispersion of nanoparticles would help for TEM beam alignment to get clear image.

Response:

We sincerely thank the reviewer for his/her excellent suggestion. As a result, we have retaken and replaced the TEM image of Fig. S9a, as shown below.

“

”

Comment 5:

How do authors obtain this equation(1)? There should be dispersion in the wafer thickness, etching orientation and SiN window size and so on, so I feel it is hard to say single nanometer levels resolution. Please check several wafer and tell the accuracy of this fitting profile.

Response:

Thank you for your valuable feedback. Equation (1) accurately represents the relationship between the spectral peak λ_p and the remaining thickness H during pre-etching. The data points for λ_p and H were obtained from tests conducted on four wafer samples. The standard deviation of the measured remaining thickness was approximately 8 nm. In response to your comment, we have made the necessary revisions to Figure 1c, as shown below (see Page 3, Line 1):

“

”

Added relative statement in the legend of Fig. 1, as below (see Page 3, Lines7-10):

“

c, The spectral peak was measured using a spectrometer, while the remaining silicon thickness was measured using a contact profilometer. The black dots represent experimental data from 4 samples, and the red line represents the fitting result. Based on four wafer tests, the standard deviation of the measured remaining silicon thickness at the tested spectral peak is approximately 8 nm.

”

Comment 6:

In addition, authors determine pore size and thickness based on optical image and TEM image. However, since TEM and optical image are not clear, it is hard to say these images can tell accurate number of pore size and thickness. To support, pore dimension, authors should add voltage vs. current data for all pore conditions. Ionic current can tell much clear pore dimension because ion current is much sensitive to pore dimension.

Response:

We thank the reviewer for the excellent and constructive suggestion.

Accordingly, we have added the voltage vs. current, i.e., I - V , curves of different nanopore samples in the Supplementary Note 9, as shown below.

“

Supplementary Note 9. TEM images and I - V curves of nanopore samples corresponding to the datapoints shown in Figure 2e and f.

Figure S8. (a-i) Back-view TEM images of MPSNs with varying remaining thickness H (or irradiation time t_i) and over-etching time t_o . (m and n) I - V curves of each MPSN shown in (a-i), measured in 1M KCl solution using Ag/AgCl electrodes. (o) Measured conductance G (extracted from the I - V curves) plotted against the measured pore sizes obtained from TEM.

”

Comment 7:

In Page 6 Line 12, authors said pore size with a standard deviation of less than 1 nm, but I could not find such resolution based on Figure 2e and f. To propose such high fabrication resolution, authors should try same fabrication condition as least 3-4 times and show the accuracy (distribution) of this pore fabrication method.

Response:

We thank the reviewer’s comment.

Accordingly, we have added a new Supplementary Note 11 to show the repeatability of the proposed fabrication method, as shown below.

“

Supplementary Note 11. The repeatability study of the photovoltaic electrochemical etch-stop technique.

Figure S10. The TEM images of nanopores fabricated using over-etching time t_o of 0.6 seconds and the samples of variant remaining thicknesses (corresponding to different measured spectral peaks), i.e., $H = 19 \pm 7$ nm and $\lambda_p = 475.8$ nm (a), $H = 76 \pm 8$ nm and $\lambda_p = 478.4$ nm (b), $H = 95 \pm 8$ nm and $\lambda_p = 479.3$ nm (c), $H = 114 \pm 8$ nm and $\lambda_p = 480.0$ nm (d).

”

We also replaced the standard deviation calculated in a single fabrication condition (i.e., $\lambda_p = 480.0$ nm, $t_o = 0.6$ seconds) with the average standard deviation of all fabrication conditions using t_o of 0.6 seconds illustrate the fabrication accuracy, as shown below (see Page 6, Lines 11-15).

“

Additionally, a repeatability study assessing the proposed MPSN fabrication method demonstrated that the measured pore sizes with an average standard deviation of about 1.1 nm could be achieved by using four identical fabrication conditions, i.e., the same over-etching time of 0.6 seconds utilized to etch three samples of the same remaining thickness (see Fig. 2e and f and Supplementary Note 11).

”

Comment 8:

In page 11 line 1, it is interesting to see that proposed system can detect molecular and particles at low concentration (0.1nM) with quite high frequency(capture rate). Authors should mention a reason why this type of pore can detect such low concentration compared with typical solid-state nanopore system which has 1-25 nm pore with 1-50 nm thickness. Maybe COMSOL simulation might help to find capture radius of this type of pore.

Response:

We appreciate the reviewer's valuable suggestion. In order to demonstrate the high capture rate and capture radius of the inertial-kinetic translocation in MPSN, we have included a new Supplementary Note 17, as presented below. It is found that the high capture rate of the inertial-kinetic translocation in MPSN is achieved as the capture radius covers the whole flow chamber.

“

Supplementary Note 17. Study the capture rate of the inertial-kinetic translocation in MPSN.

Figure S19. (a) Schematic representation of the model setup illustrating the flow cell, silicon sample, and nanopore. The centrifugal force is denoted as f_c , with the origin positioned at the center of the nanopore. (b) Random translocation traces of particle-like proteins (with a molecular weight of 50 kDa and a diameter of 5 nm) influenced by inertial force and Brownian motion in the flow chamber at a rotation speed of 4000 rpm in 1M KCl solution.

In general, a capture radius can be defined as the radial distance R' at which the diffusion-dominated dynamics cross over to the field-induced capture⁴⁴.

For electrokinetic translocation in a typical solid-state nanopore system, the molecular capture radius can be described as⁴⁴:

$$R' = \frac{QU}{kT} \frac{D_p}{8 + 2\pi} \quad (\text{S8})$$

where Q is the surface charge of the trapped molecules (such as $17e$ for the BSA nominal charge at $\text{pH} = 7$)⁴⁵, U is the applied bias (i.e., 300 mV), k is the Boltzmann constant, T is the environmental temperature, D_p is the pore size of the nanopore (i.e., 15 nm). Therefore, the capture radius of the nanopore to BSA is around 210 nm at $\text{pH} = 7$.

In the inertial-kinetic translocation, the molecular behavior is influenced by the interplay between centrifugal force and Brownian diffusion. This competition determines the capture radius by comparing the relative magnitudes of molecular inertial potential energy and thermal energy. To study this phenomenon in MPSN, a protein trace model was developed using COMSOL Multiphysics, with molecular traces determined by the Langevin equation. The centrifugal force f_c is assumed to be constant throughout the chamber. In this simulation, a protein with a molecular weight of 50 kDa and a diameter of 5 nm is released from the top of the flow chamber. The results demonstrate that the molecular translocation process is predominantly governed by inertial force, ensuring a high capture rate and capture radius throughout the entire flow chamber (see Fig S19).

”

Added new Reference 46 and 47:

“

44. Qiao, L., Ignacio, M. & Slater, G. W. Voltage-driven translocation: Defining a capture radius. *The Journal of Chemical Physics* **151** (2019).

45. Kubiak-Ossowska, K., Jachimska, B., Al Qaraghuli, M. & Mulheran, P. A. Protein interactions with negatively charged inorganic surfaces. *Current opinion in colloid & interface science* **41**, 104-117 (2019).

”

In addition, we have also added a statment about the tested molecular concentration in the main text, as shown below (see Page 8, Lines 27-31).

“

In contrast to electrokinetic translocation in a conventional solid-state nanopore system^{61, 62}, the inertial-kinetic translocation exhibits a capture radius that spans the entire upper flow chamber. This enables a significantly enhanced capture efficiency of single molecules, even at low concentrations of approximately 0.1 nM (see Supplementary Note 17).

”

Added new References 61-62 in the main text (see Page19 , Lines 2-5):

“

61 Qiao, L., Ignacio, M. & Slater, G. W. Voltage-driven translocation: Defining a capture radius. *The Journal of Chemical Physics* **151** (2019).

62 Qiao, L. & Slater, G. W. Capture of rod-like molecules by a nanopore: Defining an “orientational capture radius”. *The Journal of Chemical Physics* **152** (2020).

”

Comment 9:

In equation (2)-(5), authors show fitting result of t_1 and t_2 at 2000 and 4000rpm. My understanding is basically this sensing method involve inertial kinetic force because authors

did not find any difference between 300 and 600 mV in Figure 4. Then why can different fitting equation be applied for t_1 and t_2 ?

Response:

Thanks to the reviewer for the excellent comments.

The equations (2)-(5) are fitted to determine the relationship between dwell time (i.e., t_1 and t_2) and molecular mass (i.e., m) and shape (i.e., K') with experimental data based on the molecular motion mechanism. Here, t_1 and t_2 are the durations of capture and translocation stages, respectively. Theoretically, the molecular behaviors are governed by competition between mass-dependent centrifugal force and shape-dependent viscous diffusion in the capture stage, and the Langevin equation can be employed to elucidate the molecular motion mechanism in this stage, thereby deriving a linear correlation between t_1 and shape-to-mass weighting factors (i.e., $\xi=K'/m$), given by $t_1 = \frac{6\pi\mu R\xi L}{\rho\omega^2}$. In other words, the molecular motion in the capture stage is related to the molecular mass and shape, simultaneously. On the other hand, nanopore-particle interactions become significant and replaced viscous diffusion to compete with centrifuge force in the translocation stage, and the Eyring-like form is more suitable for explaining the molecular motion in the stage, leading to the exponential dependence of t_2 and m to be derived as $t_2 = t_0 e^{-\frac{h(m\rho\omega^2)}{kT}}$. It means the molecular motion in the translocation stage is dominated by the centrifugal force while the centrifugal force is mainly decided by the molecular mass (instead of shape) and the rotation speed. Thus, the molecular motion mechanisms determine the linear fitting relationship between t_1 and ξ and the exponential fitting relationship between t_2 and m .

Comment 10:

Do experiments in Figure 6a and b use same pore? Ration different in antibody and antibody complex can be caused by pore dimension. If not used different pore, please run experiments and show the ratio at several different pores.

Response:

Thank you to the reviewer for the valuable and constructive suggestion. The signals in Figure 6a and b were obtained from the same nanopore. To address this, we have included Supplementary Note 22, which presents the ratio of antibody and complex for both 20-nm and 23-nm MPSNs. This information is now available below.

“

Supplementary Note 22. Current traces for sensing the dissociation of antibody-antigen complex in 20- and 23-nm MPSN.

Figure S23. (a and b) Current traces obtained using 20-nm (a) and 23-nm (b) MPSNs. (1) represents the signal trace of EpCAM IgG alone, while (2-5) display the current traces of the EpCAM IgG-EpCAM antibody-antigen complex mixture measured at 2 mins, 22 mins, 42 mins, and 62 mins, respectively, after pH adjustment to 6.6 (the equilibrium point) at a rotation speed of 2000 rpm and bias voltages of 0.3 V. The antibody and antibody-antigen complex signals in the traces were labeled as "1" and "2," respectively. (c-e) Longitudinal monitoring of the dissociation dynamics of the antibody-antigen complex using 15-nm (c), 20-nm (d), and 23-nm (e) MPSNs, with four tests performed every 20 minutes.

To investigate the dissociation dynamics of the antibody-antigen complex, we conducted additional experiments using 20- and 23-nm MPSNs in addition to the 15-nm MPSN. The solution was found to have a complex ratio of $67.7\% \pm 1.8\%$, $59.6\% \pm 2.1\%$, $53.5\% \pm 3.5\%$, and $49.5\% \pm 4.3\%$ measured at 2 mins, 22 mins, 42 mins, and 62 mins, respectively (see Fig. S23). Notably, the antibody translocation signal exhibited a single peak in both the 20- and 23-nm nanopores, while the characteristic multiple peak groups were observed only in the 15-nm nanopore.

”

Comment 11:

In page 13 line 20, how authors know single, two and three peaks single, dimer and trimer nanoparticles? To make sure of this assumption, authors should prepare purified dimer or trimer nanoparticle and run the control experiment. At least, dimer nanoparticles can be prepared. See this paper: K. Esashika, R. Ishii, S. Tokihiro, and T. Saiki, "Simple and rapid method for homogeneous dimer formation of gold nanoparticles in a bulk suspension based on van der Waals interactions between alkyl chains," *Opt. Mater. Express* 9, 1667-1677 (2019).

Response:

Thank you to the reviewer for the valuable suggestion.

We attempted to purify single, dimer, and trimer Au@PEGs using the gel electrophoresis separation method described in references²⁻⁵. However, we found that the gel electrophoresis conditions (175 V for 15 min with $0.5 \times$ TBE as the running buffer) mentioned in the references were not suitable for sorting different aggregations of 4.8-nm Au@PEGs. Although this method has been proven effective for AuNPs with diameters ranging from 20 nm to 80 nm. As an alternative, we implemented a centrifugation-based separation protocol to purify aggregations of 4.8-nm spherical Au@PEGs. We have included this new protocol in a supplementary note in the supplementary information.

“

Supplementary Note 23. Current traces of Au@PEGs aggregations purified using centrifugation-based separation protocol.

Figure S24. (a) 4.8-nm spherical Au@PEGs reagents after centrifugation at rotation speeds of 14000 (1), 16000 (2), and 18000 rpm (3) using the centrifugation-based separation protocol. (b) TEM image of a mixture containing 60% dimers and 40% monomers obtained from suspensions centrifugally purified at 16000 rpm. (c) TEM image of monomers obtained from suspensions centrifugally purified at 18000 rpm. (d and e) Current traces of suspensions purified at 16000 rpm (d) and 18000 rpm (e), measured at a rotation speed of 2000 rpm and a bias voltage of 0.3 V. Insets in (d) and (e) display representative two-peak and one-peak signals, respectively.

To confirm the correspondence between single-peak signals and single molecules, as well as double-peak signals and dimers, centrifugation-based separation protocol was employed to purify different aggregations⁴⁸. The method involved systematically increasing the rotation speed to achieve stepwise g-force density centrifugation. After aggregating Au@PEGs at 92 °C, the samples were centrifuged at an initial speed of 10,000 rpm for 30 minutes. Subsequently, the rotation speed was increased by 2,000 rpm after each 30-minute centrifugation until reaching 20,000 rpm. The sediment was collected from the bottom of the centrifuge tube, and the supernatant was subjected to further centrifugation (see Fig. S24a).

At 16,000 rpm, a mixture comprising 60% dimers and 40% monomers was obtained (see Fig. S24b). Increasing the rotation speed to above 18,000 rpm resulted in almost 100% monomers (see Fig. S24c).

Furthermore, the suspensions purified at 16,000 rpm and 18,000 rpm were separately injected into the in-tube nanopore sensing system, enabling the acquisition of current blockade traces with inertial-kinetic translocation. The trace of the 16,000 rpm-purified suspensions exhibited

a ratio of two-peak signals at approximately 4:6 (see Fig. S24d), whereas all blockade signals of the 18,000 rpm-purified suspensions displayed the characteristic single peak (see Fig. S24e). This difference in ratio maps the single-peak signals to single molecules and the two-peak signals to biomolecule aggregates.

”

Newly added reference in SI:

“

48. Novak, J. P., Nickerson, C., Franzen, S. & Feldheim, D. L. Purification of molecularly bridged metal nanoparticle arrays by centrifugation and size exclusion chromatography. *Analytical chemistry* **73**, 5758-5761 (2001).

”

Reference:

2-5. Esashika, K., Ishii, R., Tokihiro, S. & Saiki, T. Simple and rapid method for homogeneous dimer formation of gold nanoparticles in a bulk suspension based on van der Waals interactions between alkyl chains. *Optical Materials Express* **9**, 1667-1677 (2019).

Comment 12:

Also there is missing explanation that why EpCAM IgG and Au@PEG nanoparticles in trimolecules aggregate showed two peaks and three peaks respectively. Authors need to refer previous works or explains detailed mechanism step by step.

Response:

Thank you to the reviewer for the valuable suggestion.

We acknowledge that previous electrokinetic studies^{2-6, 2-7} have reported the two-peak-group distribution of EpCAM IgG signals and the three-peak distribution of Au@PEG trimolecular aggregates. Notably, the bimodal characteristics of the antibody signal were significantly evident after fitting with the least squares method. Additionally, we have included a new supplementary note 19 to provide a detailed explanation of the mechanism behind the current blockade signal characteristics corresponding to EpCAM IgG and Au@PEG nanoparticles in the trimolecular aggregate.

“

Supplementary Note 21. Study the signal characteristics of EpCAM IgG and Au@PEG nanoparticles in trimolecular aggregate.

Figure S22. (a and c) Spatial distribution of the net ionic concentration difference ($v_1c_1+v_2c_2$) during the translocation of EpCAM IgG-like (a) and Au@PEG nanoparticles-like (c) trimolecular aggregates through a nanopore. Here, v_1 and v_2 represent the valences of potassium and chloride ions, respectively, while c_1 and c_2 represent the concentrations of potassium and chloride ions, respectively. (b and d) The corresponding current change ΔI associated with the EpCAM IgG-like (b) and Au@PEG nanoparticles-like (d) trimolecular aggregates.

To demonstrate the high conformational sensitivity of MPSN, we conducted simulations to observe the changes in ion concentration during the translocation of particle-like trimolecular aggregates through the sensing zone of MPSN. The simulation employed an ionic mass transport model implemented in COMSOL Multiphysics, where the mass transfer of ions and current density distribution were determined using the Nernst-Planck-Poisson (NPP) equation⁴⁷. The electrolyte used was 1M KCl, the nanopore sidewall was made of single crystal silicon, and the pore size was set at 15 nm. The simulated target molecules were EpCAM IgG-like trimolecular aggregates and Au@PEG nanoparticles-like trimolecular aggregates (see Fig. S22a and c). The corresponding feedback current signals ΔI were calculated using the conductivity equation of the electrolyte⁴⁷ (see Fig. S22b and d). The current signal of EpCAM IgG exhibited a two-peak-group characteristic, while the current signal of the Au@PEG trimolecular aggregate displayed a three-peak characteristic.

”

Newly added references in SI:

“

47 Yeh, L.-H., Zhang, M., Qian, S., Hsu, J.-P. & Tseng, S. Ion concentration polarization in polyelectrolyte-modified nanopores. *The Journal of Physical Chemistry C* **116**, 8672-8677 (2012).

”

2-6 Yusko, E. C. *et al.* Real-time shape approximation and fingerprinting of single proteins using a nanopore. *Nature nanotechnology* **12**, 360-367 (2017).

2-7 Huang, B. *et al.* Identification of plasmon-driven nanoparticle-coalescence-dominated growth of gold nanoplates through nanopore sensing. *Nature Communications* **13**, 1402 (2022).

Comment 13:

In page 13 line 28, authors proposed silicon pore with micro-thickness provide high mechanical strength to withstand inertial force. To propose the high mechanical strength, authors should show pore expansion over the time by showing current trace at certain voltage and inertial force (rpm) for several time sections such as 0, 30, 60, 120 min.

Response:

Thank you for the valuable suggestion.

In response, we have included Supplementary Note 14, which presents current traces of different time sections from different bias voltages (0.3 and 0.6 V) and rotation speeds (2000 and 4000 rpm) in order to provide further insights. These data demonstrate that there is no noticeable pore expansion effect during centrifugation.

“

Supplementary Note 14. Test the stability of MPSN.

Figure S13. Current traces were obtained using 15-nm MPSN at different bias voltages and rotation speeds: 0.3 V and 2000 rpm (a), 0.3 V and 4000 rpm (b), 0.6 V and 2000 rpm (c), and 0.6 V and 4000 rpm (d), over a duration of more than 2 hours. Only selected signal episodes starting from 0, 30, 60, and 120 minutes are shown. I_{RMS} and I_{base} represent the root-mean-square current noise and current baseline, respectively. All measurements were conducted in a 1M KCl solution.

We conducted comprehensive tests on a 15-nm nanopore, varying the bias voltages and rotation speeds. Each test lasted for over 2 hours. Our results reveal that the applied voltages affect the current baseline (I_{base}), while the rotational speed has no impact. Furthermore, the root-mean-square current noise (I_{RMS}) remains independent of both bias voltage and rotation speed. Notably, when Ag/AgCl electrodes are immersed in the KCl solution, the current baseline (I_{base}) initially drifts due to transient polarization. However, it eventually stabilizes as a steady electrochemical interface forms between the electrode surface and the electrolyte⁴². Subsequently, the I_{RMS} of the current signal consistently increases from 45 to 95 pA over a 120-minute duration, under different test conditions. This systematic change in current noise is attributed to the amplified voltage noise of the operational amplifier⁴³.

”

Newly added references in SI:

“

42 Cranny, A. & Atkinson, J. K. Thick film silver-silver chloride reference electrodes. *Measurement Science and Technology* **9**, 1557 (1998).

43 Rosenstein JK, Wanunu M, Merchant CA, Drndic M, Shepard KL. Integrated nanopore sensing platform with sub-microsecond temporal resolution. *Nature methods* **9**, 487-492 (2012).

”

Comment 14:

In Figure 5, noise level for each trace is so different. Why such huge different happen? Difference in noise level for each experiment might be influenced in analysis. Please indicate distribution of noise level at each experiment.

Response:

We appreciate the valuable suggestion provided by the reviewer.

Newly added Supplementary Note 14 demonstrates that the change in root-mean-square current noise (I_{RMS}) is primarily influenced by the amplified voltage noise of the operational amplifier. Consequently, different noise levels are observed in the representative signal traces obtained at different time sections. To mitigate the impact of noise level, we have reselected the representative blockade signals of different molecules from the same time section, specifically one hour after continuous acquisition of current signals from the in-tube nanopore system. These updated representative signals are presented in the new Figure 5. Additionally, the noise level or I_{RMS} for each experiment is indicated in the new Figure 5, as shown below.

(see Page10, Line 7):

“

Minor Comment:

1. Ref. 4 in Page1, Line 24 is wrong. It should be ref. 5.

Response:

Thanks to the reviewer. The typo Ref. 4 has been corrected to Ref. 5 (see Page1, Line 24).

“

...be integrated into portable sensing devices with electronics⁵...

”

2. In Page 1 Line 35-36, it is better to say “controlled molecular translocation”, not “controlled translocation”.

Response:

We appreciate this suggestion. The “controlled translocation” has been replaced with “controlled molecular translocation” in the manuscript (see Page 1, Line 33).

3. In Page 1 line 36, why authors only describe this paper? so far, there are so many approaches to slow down translocation speed. Here are examples;

- a. Keyser UF et al (2006) Direct force measurements on DNA in a solid-state nanopore. *Nat Phys.* 2(7):473
- b. Peng HB, Ling XS (2009) Reverse DNA translocation through a solid-state nanopore by magnetic tweezers. *Nanotechnology.* 20(18):185101
- c. Nelson EM, Li H, Timp G (2014) Direct, concurrent measurements of the forces and currents affecting DNA in a nanopore with comparable topography. *ACS Nano.* 8(6):5484–5493
- d. Hyun C et al (2013) Threading immobilized DNA molecules through a solid-state nanopore at >100 μ s per base rate. *ACS Nano.* 7(7):5892–5900
- e. Akahori, R., Yanagi, I., Goto, Y. et al. Discrimination of three types of homopolymers in single-stranded DNA with solid-state nanopores through external control of the DNA motion. *Sci Rep* 7, 9073 (2017). <https://doi.org/10.1038/s41598-017-08290-6>

Response:

Thank you.

New references 18-20, and 50 have been added accordingly. The corrections in the main text and the newly added references are shown in below.

(see Page 1, Line 33-34; Page 2, Line 9; Page 17, Lines 3-11; Page 18, Line 25-26)

“

In addition, controlled molecular translocation in nanopores has been achieved using a nanopositioner or an atomic force microscope cantilever¹⁷⁻²⁰;

18 Akahori, R. *et al.* Discrimination of three types of homopolymers in single-stranded DNA with solid-state nanopores through external control of the DNA motion. *Scientific reports* 7, 9073 (2017).

19 Hyun, C., Kaur, H., Rollings, R., Xiao, M. & Li, J. Threading immobilized DNA molecules through a solid-state nanopore at > 100 μ s per base rate. *Acs Nano* 7, 5892-5900 (2013).

20 Nelson, E. M., Li, H. & Timp, G. Direct, concurrent measurements of the forces and currents affecting DNA in a nanopore with comparable topography. *ACS nano* 8, 5484-5493 (2014).

”

“

slowing the translocation speed by introducing new potential gradients⁴⁵⁻⁵⁰, ...

50. Keyser UF et al (2006) Direct force measurements on DNA in a solid-state nanopore. *Nat Phys.* 2(7):473

”

4. There is no axis descriptions on graph in SI note 6. Please add it otherwise I cannot understand what it is saying.

Response:

Thanks to the reviewer for pointing out this mistake. The axis descriptions have been added to the Figure S5 in the updated SI note 6:

“

Figure S5. (a) Experimental setup schematic for studying the change in remaining thickness (H) during the pre-etching process. The transmission spectral peak (λ_p) blue-shifted to 520 nm, and nanopore's back-side structure profiles were measured every 300 seconds using a profilometer (Tencor Alpha-Step 500 profilometer, KLA-Tencor). The corresponding λ_p values (520, 512, 503, and 493 nm for 0, 300, 600, and 900 seconds) were recorded. (b) Profiles of the front side (black) and back side of MPSN at $\lambda_p = 520$ (gold), 512 (blue), 503 (cyan), and 493 (red) nm in the area A shown in (a), as measured by the profilometer. The origin is set on the backside surface of MPSN at $\lambda_p = 520$ nm, and the Z-axis represents the height along a cross section of the X-axis.

”

5. In page 8 line 14, this sentence is incorrect. I agree with elongated sensing length provide longer dwell time. However, the elongated sensing length does not provide better temporal resolution. If authors want to propose this concept, they should add additional explanation for it.

Response:

We would like to thank the reviewer for this constructive comment. Accordingly, we updated our statement.

Original statement:

“

Additionally, the deep pyramidal structure of the MPSNs (approximately 21.25- μm deep) offers an elongated sensing length supporting longer dwell times and better temporal resolutions.

”

Revised text (see Page 8, Lines 16-17):

“

Additionally, the deep pyramidal structure of the MPSNs (approximately 21.25- μm deep) offers an elongated sensing length supporting longer dwell times.

”

6. In Figure 4, authors should add events number in all histogram to show better statistically understanding. Also please add concentration of Au@PEG and EpCAM.

Response:

Thanks to the reviewer for the suggestion.

Accordingly, we have added events number (N) in histograms and the concentration information (C_m) of Au@PEG and EpCAM, as shown below (see Page 9, Line 1).

“

”

In addition, we have also added a statement about the tested molecular concentrations in the figure legend, as below (Page 9, Lines 3-7; Page 9, Lines 9-11):

“

a, Dwell time histograms of Au@PEG measured using electrokinetic translocation (blue) versus inertial-kinetic translocation (red) in a 15-nm MPSN with fitted Gaussian distribution curves. Electrokinetic translocation of Au@PEG with molecular concentration C_m of 20 nM was performed under voltage biases U of 0.3 V (I) and 0.6 V (II) when pH is 7.0, while inertial-kinetic translocation of Au@PEG (III-VI) with C_m of 0.2 nM was conducted under different rotation speeds ω with a pH of 3.0.

Panels (e) and (f) show the similar study as displayed in (a) and (b) to demonstrate the long and stable dwell times achieved using inertial-kinetic translocation of EpCAM in MPSN; with $C_m = 20$ and 0.2 nM for electrokinetic and inertial translocation, respectively; N indicates translocation event number in all histogram.

”

Reviewer #3 (Remarks to the Author):

The author carefully addressed the comments from the reviewers. There were issues with the references and claims especially in the introduction part, and the author have addressed most of them. The manuscript is much better now and can be published.

I still recommend that the authors show the detection of DNA using their setup, which is an essential work to demonstrate the capability of the method. This could be done in future work. Some minor comments are listed below before the publication.

1) Table S1. Please cite the reference directly and avoid using the indirect citation in Table S1 (ref 18), with regards to the “Glass pulling” and “Chemical etching”. Where are the reference for glass nanopores?

The “Glass pulling” reference should also be cited in Page 13, Line 33: “fabricated by using glass pulling and silicon chemical etching techniques”. Reference are missing here.

2) Page 13, Line 35. “conventional glass pulling techniques still need further optimization to consistently produce nanopores of same size and conventional chemical etching techniques are not able to reliably fabricate silicon nanopores smaller than 8 nm” The authors still have dangerous claims without checking enough references on nanopore research. First, Ulrich F. Keyser’s group has shown that the pore size is pretty consistent with their typical pore size of 14 ± 3 nm. Second, “chemical etching techniques are not able to reliably fabricate silicon nanopores smaller than 8 nm”. Please add references here. Please also thoroughly check other references in the manuscript.

Response:

We appreciate the reviewer's recommendation for publication.

For the remaining two minor comments, the responses are provided below.

1) We have replaced the indirect citation (ref 18) with direct citations (refs 24-31) in the Table S1, see below.

24 Lan, W.-J., Holden, D. A., Zhang, B. & White, H. S. Nanoparticle transport in conical-shaped nanopores. *Analytical chemistry* **83**, 3840-3847 (2011).

25 Steinbock, L. J., Bulushev, R. D., Krishnan, S., Raillon, C. & Radenovic, A. DNA translocation through low-noise glass nanopores. *Acs Nano* **7**, 11255-11262 (2013).

26 Chen, K. *et al.* Super-Resolution Detection of DNA Nanostructures Using a Nanopore. *Advanced Materials* **35**, 2207434 (2023).

27 Holden, D. A., Hendrickson, G., Lyon, L. A. & White, H. S. Resistive pulse analysis of microgel deformation during nanopore translocation. *The journal of physical chemistry C* **115**, 2999-3004 (2011).

28 Alawami, M. F. *et al.* Lifetime of glass nanopores in a PDMS chip for single-molecule sensing. *Iscience* **25** (2022).

29 Deng, T., Chen, J., Wu, C. & Liu, Z. Fabrication of inverted-pyramid silicon nanopore arrays with three-step wet etching. *ECS journal of solid state science and technology* **2**, P419 (2013).

30 Park SR, Peng H, Ling XS. Fabrication of nanopores in silicon chips using feedback chemical etching. *Small* **3**, 116-119 (2007).

31 Chen Q, Wang Y, Deng T, Liu Z. Fabrication of nanopores and nanoslits with feature sizes down to 5 nm by wet etching method. *Nanotechnology* **29**, 085301 (2018).

And we have added citations (i.e., refs 36 and 66) for supporting the discussion in the revised manuscript, see below (see Page 13, Lines 40-42; Page 17, Lines 45-46; Page 19, Lines 12-14).

A solid-state funnel-shaped nanopore of over tens micrometres thickness can meet these requirements, and such a nanopore can be fabricated by using glass pulling and silicon chemical etching techniques^{36, 66}.

36. Park SR, Peng H, Ling XS. Fabrication of nanopores in silicon chips using feedback chemical etching. *Small* **3**, 116-119 (2007).

66. Holden, D. A., Hendrickson, G., Lyon, L. A. & White, H. S. Resistive pulse analysis of microgel deformation during nanopore translocation. *The journal of physical chemistry C* **115**, 2999-3004 (2011).

2) We have revised the claims about the glass pulling technologies and added citations (refs 37-39) to support the discussion about the conventional chemical etching techniques in Page 13, Line 35 in the revised manuscript, see below (see Page 13, Lines 42-44; Page 17, Lines 47-50; Page 18, Lines 1-2).

However, conventional chemical etching techniques are not able to reliably fabricate silicon nanopores smaller than 8 nm³⁷⁻³⁹.

37. Chen Q, Wang Y, Deng T, Liu Z. Fabrication of nanopores and nanoslits with feature sizes down to 5 nm by wet etching method. *Nanotechnology* **29**, 085301 (2018).

38. Chen Q, Liu Z. Fabrication and applications of solid-state nanopores. *Sensors* **19**, 1886 (2019).

39. Liu H, Zhou Q, Wang W, Fang F, Zhang J. Solid-State Nanopore Array: Manufacturing and Applications. *Small* **19**, 2205680 (2023).

In addition, we have also thoroughly checked other references in the revised manuscript.

REVIEWER COMMENTS

Reviewer #1 (Remarks to the Author):

All my comments have been addressed, and I recommend the manuscript for publication

Reviewer #2 (Remarks to the Author):

In this paper, Yang et al. developed fabrication method of crystal silicon nanopore using photoinhibition effect and demonstrate protein and Au nanoparticle detection using inertial force. In revised manuscript, author made tremendous revisions and added additional experimental data to support their work. Mostly I am getting close satisfied with this revised manuscript but I still have a few more comments for current version.

My comments are the following;

Major Comments:

1. In Figure S8, author added new IV curves which make clear to understand geometry of pore. I can see the trend that current increase with pore size. However, it seems there is no effect of membrane thickness on current. For example, 66.0nm (t=5s, H=19nm) and 64.6nm (t=5s, H=114nm) looks almost same conductance. Typically, pore conductance is explained as following equation, so conductance is reduced 8-9 time in general sense. Why this pore does not follows this equation? For info of equation please check this paper (https://bpb-us-w2.wpmucdn.com/web.sas.upenn.edu/dist/9/448/files/2018/04/wanunu_nnano_2010-2i5pb9s.pdf).

$$I = \sigma_s \Delta V \left(\frac{4h}{\pi d^2} + \frac{1}{d} \right)^{-1}$$

2. This pyramidal silicon nanopore shows quite long dwell time compared with nanopore in 10-50nm thick membranes. Typically, dwell time of pore made in nanometer thick silicon nitride membrane is less than 1 msec (<https://pubs.acs.org/doi/10.1021/ac4012045>). I assume this difference happen due to sensing area is large in this case. Please add description of where current drop start using electric field profile obtained by COMSOL.

3. Supplementary Note 17 shows COMSOL simulation that inertial force can capture molecules at far distance from a pore compared with electro-kinetical force. This is valuable analysis, but it is better to

have COMSOL simulation for electro-kinetical force. Comparison of these two-different driving force makes much convicting for author's argument.

Minor Comment:

1. Fig. S11 c show flipped IV curve which shows negative current at positive voltage conditions. Normally positive voltage shows positive current. Please correct this or add explanation for this IV curve.

2. In table S1, "TEM shrinking" is incorrect because shrinking is used optionally after pore was fabrication. Term should be corrected such as "TEM drilling."

3. Also thickness in TEM says "Not decided by techniques". I do not think this is correct because TEM drilling does not work if membrane is too thick.

Reviewer #3 (Remarks to the Author):

The authors have well addressed all my comments. Overall, the paper provides an insightful mechanism to regulate the translocation of molecules through nanopore. After revision, the experiment and analysis procedures are detailed and reliable. Therefore, I believe the manuscript can be accepted in Nature Communications.

Responses to the reviewers' comments

We are grateful to the reviewers' comments and recommendation for publication. We have revised the manuscript accordingly and the added or changed parts are marked in **red** in the manuscript text file. We have also provided the detailed responses as appended below. Author responses are in **blue**, changes and additions to the manuscript are in **red**, original texts in manuscript and reviewer's comments are in **black**, and all line numbers refer to the documents with redlined highlights.

Reviewer #1 (Remarks to the Author):

All my comments have been addressed, and I recommend the manuscript for publication.

Response:

We wish to thank the reviewer for the valuable feedback and recommendation all along during the review process.

Reviewer #2 (Remarks to the Author):

In this paper, Yang et al. developed fabrication method of crystal silicon nanopore using photoinhibition effect and demonstrate protein and Au nanoparticle detection using inertial force. In revised manuscript, author made tremendous revisions and added additional experimental data to support their work. Mostly I am getting close satisfied with this revised manuscript, but I still have a few more comments for current version.

Response:

We would like to express our sincere gratitude to the reviewer for his/her positive feedback and invaluable comments.

Major Comments:

1. In Figure S8, author added new IV curves which make clear to understand geometry of pore. I can see the trend that current increase with pore size. However, it seems there is no effect of membrane thickness on current. For example, 66.0nm (t=5s, H=19nm) and 64.6nm (t=5s, H=114nm) looks almost same conductance. Typically, pore conductance is explained as following equation, so conductance is reduced 8-9 time in general sense. Why this pore does not follows this equation? For info of equation please check this paper (https://bpb-us-w2.wpmucdn.com/web.sas.upenn.edu/dist/9/448/files/2018/04/wanunu_nnano_2010-2i5pb9s.pdf).

$$I = \sigma_s \Delta V \left(\frac{4h_{eff}}{\pi d^2} + \frac{1}{d} \right)^{-1} \quad (1)$$

Response:

Figure R1. Schematic diagrams of a typical sub-60-nm-thick silicon-nitride nanopore (a) and a MPSN (b), illustrating the different effective pore thickness h_{eff} in respective nanopore. Here, h is the membrane thickness; d is the diameter of nanopore; θ is 54.7° ; H is the remaining thickness.

We thank the reviewer for the comment.

As shown in Fig. R1a, for the sub-60-nm-thick silicon-nitride nanopores, the effective pore thickness, h_{eff} , in Equation (1) is defined as the reduced thickness of an equivalent cylinder, which can be described as²⁻¹:

$$h_{eff} = \frac{h}{3.04 \pm 0.30} \quad (2)$$

Therefore, the h_{eff} and resultant conductance ($G = \frac{I}{\Delta V}$) of silicon nitride nanopore is directly correlated to the membrane thickness, h .

By substituting Equation (2) into Equation (1), the conductance G of the typical silicon-nitride nanopores can be described as:

$$G = \sigma \left(\frac{1.32h}{\pi d^2} + \frac{1}{d} \right)^{-1} \quad (3)$$

Therefore, the conductance of the silicon-nitride nanopores is related to the nanopore diameter d , the membrane thickness h , and the solution conductivity σ .

As for the micropillar silicon nanopore (MPSN) fabricated in our study, the effective pore thickness h_{eff} is described as²⁻²:

$$h_{eff} = 0.92 \sqrt{\frac{90}{90 - \theta}} d \quad (\text{for pyramidal pores with } h_{eff} < h) \quad (4)$$

Therefore, the h_{eff} of MPSN depends on the nanopore diameter d and the slanted sidewall angle θ , but not depends on the membrane thickness h and the remaining thickness H (see Fig. R1b).

By substituting Equation (4) into Equation (1), the conductance G of the MPSN can be described as²⁻²:

$$G = \frac{\pi d \sigma}{3.68} \sqrt{\frac{90 - \theta}{90}} \quad (5)$$

Therefore, the conductance of MPSN is related to the nanopore diameter d and the solution conductivity σ , but independent of the membrane thickness h and the remaining thickness H .

To clarify the conductance of MPSN, we have added the related equation into Supplementary Note 9.

“

Figure S8. (a-l) Back-view TEM images of MPSNs with varying remaining thickness H (or irradiation time t_i) and over-etching time t_o . (m and n) I - V curves of each MPSN shown in (a-l), measured in 1M KCl solution using Ag/AgCl electrodes. (o) Measured conductance G (extracted from the I - V curves) plotted against the measured pore sizes obtained from TEM. Notably, the conductance follows the equation $G = \frac{\pi d \sigma}{3.68} \sqrt{\frac{90-\theta}{90}}^{40,41}$, where d , σ , and θ are the pore size of nanopore, the solution conductivity, and the slanted sidewall angle of 54.7° , respectively.

”

Newly added references in SI:

“

- 40 Wanunu, M. et al. Rapid electronic detection of probe-specific microRNAs using thin nanopore sensors. *Nature Nanotechnology* **5**, 3149-3153 (2010).
- 41 Wen, C., Zhang, Z. & Zhang, S.-L. Physical Model for Rapid and Accurate Determination of Nanopore Size via Conductance Measurement. *ACS Sensors* **2**, 1523-1530 (2017).

”

References of current response:

- 2-1 Wanunu, M. et al. Rapid electronic detection of probe-specific microRNAs using thin nanopore sensors. *Nature Nanotechnology* **5**, 3149-3153 (2010).
- 2-2 Wen, C., Zhang, Z. & Zhang, S.-L. Physical Model for Rapid and Accurate Determination of Nanopore Size via Conductance Measurement. *ACS Sensors* **2**, 1523-1530 (2017).

2. This pyramidal silicon nanopore shows quite long dwell time compared with nanopore in 10-50nm thick membranes. Typically, dwell time of pore made in nanometer thick silicon nitride membrane is less than 1 msec (<https://pubs.acs.org/doi/10.1021/ac4012045>). I assume this difference happen due to sensing area is large in this case. Please add description of where current drop start using electric field profile obtained by COMSOL.

Response:

Thank you for the constructive suggestion. Accordingly, we have added the description of sensing area of MPSN and where the current drop starts and ends in the sensing area into Supplementary Note 15.

“

Supplementary Note 15. Sensing area of MPSN.

Figure S14. (a) The distribution of electric field in MPSN when the bias voltage is 0.3 V. The inset shows an enlarged view around the nanopore. The sensitive length L along the z direction is defined as the distance between the two points at which the electric field intensity is decayed to e^{-1} of the maximum intensity located in the nanopore centre. (b-e) Simulated translocation of a 7-nm circular molecule with surface charge of $17e$ through different locations at $z = 85$ (b), 45 (c), 5 (d), and -15 nm (e) by calculating spatial distribution of the net ionic concentration difference ($v_1c_1+v_2c_2$) in the MPSN within the area defined by x from -200 nm to 200 nm and z from -100 nm to 200 nm. Here, v_1 and v_2 represent the valences of potassium and chloride ions, respectively, while c_1 and c_2 represent the concentrations of potassium and chloride ions, respectively. (f) The corresponding current blockade change ΔI caused by the molecular translocation. The x - and z -direction are set to parallel to $[0,1,0]$ and $[1,0,0]$ crystal orientation of the silicon samples, respectively. The origin is positioned at the centre of the nanopore.

By implementing an ionic mass transport model in COMSOL, simulations were conducted to study the ion concentration change during the translocation of a circular molecule and therefore deduce the sensing area and length of MPSN (see Fig. S14a). The distribution of electrical field and the net ionic concentration were calculated using the Helmholtz Equation and Nernst-Planck-Poisson (NPP) equation, respectively⁴⁸. In the simulation, the electrolyte was 1M KCl, the nanopore material was single crystal silicon, and the pore size was set at 15 nm. The current signals were calculated using the conductivity equation of the electrolyte⁴⁸, which can be described as:

$$I = \int_S F(v_1\vec{N}_1 + v_2\vec{N}_2) \cdot \vec{n}dS \quad (S9)$$

where F is the Faraday constant; v_1 and v_2 represent the valences of potassium and chloride ions, respectively, while \vec{N}_1 and \vec{N}_2 represent the ionic flux of potassium and chloride ions, respectively; S is surface of the sensing area of the MPSN; \vec{n} is outward unit normal vector. By simulating the translocation of a 7-nm circular molecule with a surface charge of $17e$ through a nanopore, the study reveals that the current drop begins when the molecule's centre is positioned at $z = 100$ nm, and ends at -15 nm (see Fig. S14b-f).

”

Newly added references in SI:

“

48 Yeh, L.-H., Zhang, M., Qian, S., Hsu, J.-P. & Tseng, S. Ion concentration polarization in polyelectrolyte-modified nanopores. *The Journal of Physical Chemistry C* **116**, 8672-8677 (2012).

”

3. Supplementary Note 17 shows COMSOL simulation that inertial force can capture molecules at far distance from a pore compared with electro-kinetical force. This is valuable analysis, but it is better to have COMSOL simulation for electro-kinetical force. Comparison of these two-different driving force makes much convicting for author's argument.

Response:

We thank the reviewer for this excellent suggestion. Accordingly, we have added the COMSOL simulation of electrokinetic translocation and electro-kinetic force into Supplementary Note 17.

“

Supplementary Note 17. Compare the capture radius of the inertial-kinetic translocation with electrokinetic translocation in MPSN.

Figure S19. (a) Schematic of the model setup consisting of flow chamber, silicon sample, and nanopore. The upper boundary and the nanopore are set to open. The centrifugal force f_c and the electrophoretic force f_{EP} directed towards nanopores dominates the molecular behaviours in the inertial-kinetic translocation and the electrokinetic translocation in MPSN, respectively. The x- and z-direction are set to parallel to $[0,1,0]$ and $[1,0,0]$ crystal orientation of the silicon samples, respectively. The origin is positioned at the centre of the nanopore. Simulated translocation traces and capture radius of particle-like proteins in MPSN operated (b) under inertial force f_c and Brownian motion at a rotation speed of 4000 rpm and (c) under electrokinetic force f_{EP} and Brownian motion in 1M KCl solution. Red dots in (b) and (c) indicate the original locations of proteins with a molecular weight of 50 kDa, a diameter of 5 nm, and surface charge of $17e$. f_{cmax} and f_{EPmax} denote the maximum centrifugal force and the maximum electrophoretic force, respectively. (d) and (e) show the dependence of molecular capture possibility on the molecule-nanopore distance under the inertial-kinetic translocation (d) and electrokinetic translocation (e). Here, the molecular capture possibility is defined as the ratio of molecular translocation events to all the molecular motion events, the molecule-nanopore distance is the length between the original location of the molecule and the nanopore centre, the capture radius is defined as the radial distance R' where the electrokinetic or the inertial-kinetic force starts to surpasses the diffusion-dominated dynamics⁴⁹. In general, the molecular capture possibility is 100% while molecule-nanopore distance is within the capture radius R' of nanopore.

For electrokinetic translocation in a typical solid-state nanopore system, the molecular capture radius can be described as⁴⁹:

$$R' = \frac{QU}{kT} \frac{D_p}{8 + 2\pi} \quad (\text{S10})$$

where Q is the surface charge of the trapped molecules (such as $17e$ for the BSA nominal charge at $\text{pH} = 7$)⁵⁰, U is the applied bias (i.e., 300 mV), k is the Boltzmann constant, T is the

environmental temperature, D_p is the pore size of the nanopore (i.e., 15 nm). Therefore, the capture radius of the nanopore to BSA is around 210 nm at pH =7.

For the inertial-kinetic translocation in the MPSN, the molecular behaviour is influenced by the interplay between centrifugal force and Brownian diffusion. This competition determines the capture radius R' by comparing the relative magnitudes of molecular inertial potential energy and thermal energy. On the other hand, for the electrokinetic translocation in MPSN, the molecular capture radius R' can be determined by comparing the relative magnitudes of molecular electrophoretic force and local friction drag, and therefore the capture radius can be described as⁴⁹:

$$R' = \frac{kT}{QE(R')} \quad (S11)$$

where $E(R')$ is the electric field intensity at the capture radius R' .

To compare the capture radius (R') of two translocation methods in MPSN, a molecular motion model was developed using COMSOL Multiphysics. The model utilized the Langevin equation to study molecular traces. The simulation area ranged from $x = -25 \mu\text{m}$ to $25 \mu\text{m}$ and $z = 0 \mu\text{m}$ to $60 \mu\text{m}$ (see Fig S19a). A simulated protein with a molecular weight of 50 kDa, a diameter of 5 nm, and a surface charge of 17e was released within the simulated area (see Fig S19b and c). The simulation concluded when the protein exited the simulation area through translocation. In the inertial-kinetic translocation simulation, the centrifugal force f_c is assumed to be constant throughout the chamber ($f_c/f_{cmax} = 1$) (see Fig S19b). The results demonstrate that the molecular translocation process is predominantly governed by inertial force, ensuring the proteins released within the pyramid structure with 20- μm thickness will be translocated through the nanopore. Thus, the capture radius R' of the inertial-kinetic translocation in the MPSN is around 20 μm ⁴⁹.

As for the electrokinetic translocation, the electrophoretic force f_{EP} is calculated through simulating the distribution of electric field intensity E in MPSN (see Fig S19c). The results show that the molecular capture possibility is 100% when the distance between the starting location of molecule and the nanopore centre is within the capture radius, i.e., 350 nm, where the molecular behavior is dominated by the electrokinetic force. Compared with the electrokinetic translocation, the inertial-kinetic translocation provides a longer capture radius in MPSN (see Fig S19d and e).

”

Newly added references in SI:

“

49 Qiao, L., Ignacio, M. & Slater, G. W. Voltage-driven translocation: Defining a capture radius. *The Journal of chemical physics* **151** **24**, 244902 (2019).

”

Minor Comment:

1. Fig. S11 c show flipped IV curve which shows negative current at positive voltage conditions. Normally positive voltage shows positive current. Please correct this or add explanation for this IV curve.

Response:

Thanks the reviewer.

Accordingly, we have revised the Figure S11 to indicate the input current I_{IN} and the output voltage V_{OUT} , as shown below:

“

Supplementary Note 12. Setting of the in-tube nanopore sensing device.

Figure S11. (a) Image (left) of in-tube nanopore sensing device, including a dominant tube and a battery pack. Schematic diagram (right) of internal detection modules in dominant tube. Inset within red dotted line shows the components of the flow chamber assembly, including Ag/AgCl electrodes, O-rings, coppers and the MPSN sample. (b) Simplified circuit diagram of the device. I_{IN} indicates the input current of the operational amplification circuit from current response of MPSN under applying potential V_{DC} . V_{OUT} is the output voltage of the operational amplification circuit. (c) The gain curve (V_{OUT} - I_{IN} curve) of the operational amplification circuit in (b).

Fig. S11c shows the gain curve between the output voltage V_{OUT} and the input current I_{IN} of the operational amplification circuit in Fig. S11b, which is used to amplify the blockade current signal. As the positive potential V_{DC} is connected to the negative input terminal of the operational transresistance amplifier, the gain characteristics can be described as:

$$V_{OUT} = -I_{IN}R_F \quad (S7)$$

where R_F is the transresistance.

”

2 and 3. In table S1, “TEM shrinking” is incorrect because shrinking is used optionally after pore was fabrication. Term should be corrected such as “TEM drilling”. Also, thickness in TEM says “Not decided by techniques”. I do not think this is correct because TEM drilling does not work if membrane is too thick.

Response:

We thanks the reviewer for pointing out the mistakes.

Accordingly, we have replaced “TEM shrinking” with “TEM drilling”. Furthermore, the thickness limitations of respective materials have been amended to replace “Not decided by techniques” in the updated Table S1:

“

Table S1. Conventional nanopore fabrication techniques versus PALE.

Fabrication Methods	Material	Thickness	Pore size	Pore shape	Scalability/repeatability
FIB	SiN ¹⁻⁵	< 30 nm	[0.3 nm, 280 nm] ^{1,2,6-8}	Cylindrical	Yes
	SiO ₂ ^{8,9}	[0.5 nm, 60 nm]		Cylindrical	
	SiC ^{6,10}	< 20 nm ⁶		Cylindrical	

	2D materials ^{7,11,12}	around 0.3nm ¹³		Cylindrical	
TEM drilling	SiN	< 40 nm ¹⁴	> 2 nm ¹⁴⁻¹⁸	Cylindrical	Yes
	MoS ₂	around 0.7 nm ^{17,18}		Cylindrical	
	SiO ₂	< 40 nm ¹⁴		Cylindrical	
E-beam lithography	Si	50-100 nm ^{19,20}	5 -113 nm	Funnel-shaped	Yes
Controlled dielectric breakdown	SiN	>10nm ²¹	> 1.1 nm ²¹⁻²⁵	Cylindrical	Yes
	SiO ₂	>30nm ²²		Cylindrical	
	HfO ₂	>10nm ²³		Cylindrical	
Glass pulling	glass ²⁶⁻³⁰	nanopore with over tens micrometers ²⁹	>11 nm ³⁰	Funnel-shaped	Yes
Chemical etching	Si ³¹⁻³³	nanopore with micro thick membrane ³²	> 8 nm ³³	Funnel-shaped	Yes
Photovoltaic electrochemical etch-stop technique (our method) ^{34,35}	Si	over tens micrometers	> 4.5 nm	Funnel-shaped	Yes

”

Newly added references in SI:

“

- 14 Wu, M.-Y., Krapf, D., Zandbergen, M. W., Zandbergen, H. W. & Batson, P. E. Formation of nanopores in a SiN/SiO₂ membrane with an electron beam. *Applied Physics Letters* **87**, 113106 (2005).
- 17 Graf, M. *et al.* Fabrication and practical applications of molybdenum disulfide nanopores. *Nature Protocols* **14**, 1130-1168 (2019).
- 18 Feng, J. *et al.* Single-layer MoS₂ nanopores as nanopower generators. *Nature* **536**, 197-200 (2016).

”

Reviewer #3 (Remarks to the Author):

The authors have well addressed all my comments. Overall, the paper provides an insightful mechanism to regulate the translocation of molecules through nanopore. After revision, the experiment and analysis procedures are detailed and reliable. Therefore, I believe the manuscript can be accepted in Nature Communications.

Response:

We wish to thank the reviewer for the valuable feedback and recommendation all along during the review process.

REVIEWERS' COMMENTS

Reviewer #2 (Remarks to the Author):

In this paper, Yang et al. developed fabrication method of crystal silicon nanopore using photoinhibition effect and demonstrate protein and Au nanoparticle detection using inertial force. In revised manuscript, author response properly and, I am mostly satisfied with this revised manuscript but I still have a few comments for current version to be better paper.

My comments are the following:

Major Comments:

In response of my questions, author construct an equation which can estimate pore conductance for micro-pyramidal silicon nanopore. If this proposed equation is applicable for this type of pore, author should fit with experimental data in Figure S8 m and n using this equation.

Minor Comment:

In table S1, author correct pore size of TEM drilling to be > 2 nm. This is incorrect. Please check this paper. <https://www.nature.com/articles/s41467-019-10265-2>